# A revised road map for the commitment of human cord blood CD34-negative hematopoietic stem cells

Keisuke Sumide [1], Yoshikazu Matsuoka[1], Hiroshi Kawamura[1,2], Ryusuke Nakatsuka[1], Tatsuya Fujioka[1], Hiroaki Asano[3], Yoshihiro Takihara[4,5] & Yoshiaki Sonoda[1]

We previously identified CD34-negative (CD34−) severe combined immunodeficiency (SCID)-repopulating cells as primitive hematopoietic stem cells (HSCs) in human cord blood. In this study, we develop a prospective ultra-high-resolution purification method by applying two positive markers, CD133 and GPI-80. Using this method, we succeed in purifying single long-term repopulating CD34− HSCs with self-renewing capability residing at the apex of the human HSC hierarchy from cord blood, as evidenced by a single-cell-initiated serial trans-plantation analysis. The gene expression profiles of individual CD34+ and CD34− HSCs and a global gene expression analysis demonstrate the unique molecular signature of CD34− HSCs. We find that the purified CD34− HSCs show a potent megakaryocyte/erythrocyte differ-entiation potential in vitro and in vivo. Megakaryocyte/erythrocyte progenitors may thus be generated directly via a bypass route from the CD34− HSCs. Based on these data, we propose a revised road map for the commitment of human CD34− HSCs in cord blood.

[1] Department of Stem Cell Biology and Regenerative Medicine, Graduate School of Medical Science, Kansai Medical University, Hirakata 573-1010 Osaka, Japan. [2] Department of Orthopedic Surgery, Kansai Medical University, Hirakata 573-1010 Osaka, Japan. [3] School of Nursing, Kyoto Prefectural University of Medicine, Kyoto 602-8566 Kyoto, Japan. [4] Department of Stem Cell Biology, Research Institute for Radiation Biology and Medicine, Hiroshima University, Hiroshima 734-8553 Hiroshima, Japan. [5] Japanese Red Cross Osaka Blood Center, Osaka 536-0025 Osaka, Japan. Correspondence and requests for materials should be addressed to Y.S. (email: sonoda@hirakata.kmu.ac.jp)

Hematopoietic stem cells (HSCs) are a self-renewing population with the developmental potential to give rise to all types of mature blood cells[1–3]. It is well-documented that HSCs possess enormous therapeutic potential in the context of hematopoietic stem cell transplantation (HSCT) and regenerative medicine[4–7]. Recent advances in fluorescence-activated cell sorting (FACS) technology have enabled prospective isolation of murine HSCs to high purity using various cell surface markers, including CD34, Sca-1 and the SLAM family receptors[8,9]. Among them, the CD34 antigen has long been believed to be a reliable HSC marker in mammals[10]. Two decades ago, Nakauchi et al., however, challenged this long-standing dogma, showing that murine long-term (LT) lympho-myeloid reconstituting HSCs (LT-HSCs) are lineage negative (Lin⁻), c-kit-positive (c-kit⁺), Sca-1-positive (Sca-1⁺) and CD34⁻low/negative (CD34$^{low/-}$) (CD34$^{low/-}$ KSL) cells[11]. Furthermore, individual purified CD34$^{low/-}$ KSL cells were able to fully reconstitute lympho-myeloid hematopoiesis in recipient mice.

In contrast, the purification and isolation of bona fide human CD34⁻ HSCs has lagged far behind the abovementioned murine CD34$^{low/-}$ KSL cells[11]. However, a number of studies have suggested that human bone marrow (BM)-derived and cord blood (CB)-derived CD34$^{low/-}$ cell populations contain LT-HSCs[12–14]. Dick et al. developed a SCID-repopulating cell (SRC) assay to measure primitive human HSCs in a xenotransplantation setting with NOD/SCID mice[15,16]. Using this system, Bhatia et al. first reported that SRCs are present in human BM-derived and CB-derived Lin⁻CD34⁻ cells[17]. However, the incidence of SRCs in Lin⁻CD34⁻ cells was reportedly very low (1/125,000).

We previously identified very primitive CD34⁻ SRCs in human CB using the intra-bone marrow injection (IBMI) method[18] and proposed a new concept for the hierarchy in the human HSC compartment[19,20]. However, the incidence of CD34⁻ SRC in 13 Lin⁻CD34⁻ cells (1/25,000) was still low[18]. We then developed a high-resolution purification method capable of enriching CD34⁻ SRCs at a 1/1000 level in an 18Lin⁻CD34⁻ fraction[21]. In addition, we further identified CD133 as a positive marker for CD34⁻ as well as CD34⁺ SRCs[22], which can enrich CD34⁺ and CD34⁻SRCs at approximately 1/100 and 1/140 in 18Lin⁻CD34$^{+/-}$CD133⁺ fractions, respectively[20,22]. Very recently, we demonstrated that the glycosylphosphatidylinositol-anchored protein GPI-80, which was originally reported to regulate neutrophil adherence and migration[23,24], was also expressed on human full-term CB-derived 18Lin⁻CD34⁺CD38⁻ and 18Lin⁻CD34⁻ cells[25]. Interestingly, CB-derived CD34⁻ SRCs were highly enriched in the 18Lin⁻CD34⁻GPI-80⁺ cell fraction at the 1/20 level[25].

In this study, we combine two positive/enrichment markers, CD133 and GPI-80, in order to achieve ultra-high purification of CD34⁺ and CD34⁻HSCs and successfully purify both SRCs at 1/5 and 1/8 cell levels, each of which turns out to be the highest purification levels to date. We then explore the biological nature of human CB-derived CD34⁺ and CD34⁻SRCs (HSCs) to clarify the difference in their stem cell nature using single-cell-based in vivo transplantation and gene expression analyses. These detailed single-cell-based analyses allow us to distinguish human CB-derived CD34⁺ and CD34⁻HSCs and map CD34⁻ HSCs at the apex of the human HSC hierarchy.

## Results

### Development of an ultra-high-resolution purification method.
Using two positive markers CD133[22] and GPI-80[25], we developed an ultra-high-resolution purification method for isolating CD34⁺ and CD34⁻HSCs at the single-cell level (Fig. 1a–f). The 18Lin⁻CD34⁺CD38⁻CD133⁺GPI-80$^{+/-}$ (R6 and R7) (abbreviated as 34⁺38⁻133⁺80$^{+/-}$) cells and the 18Lin⁻CD34⁻CD133⁺GPI-80$^{+/-}$ (R8 and R9) (abbreviated as 34⁻133⁺80$^{+/-}$) cells were sorted for subsequent in vitro and in vivo experiments. Photomicrographs of the purified 34⁺38⁻133⁺80$^{+/-}$ and 34⁻133⁺80$^{+/-}$ cells are shown in Fig. 1g. All cells showed immature blast-like morphologies. The area of the 34⁺38⁻133⁺80$^{+/-}$ cells was significantly larger than that of the 34⁻133⁺80$^{+/-}$ cells (Fig. 1h).

In a separate set of experiment, we analyzed the expression patterns of CD90, CD49f, CD93 and CXCR4 on the surfaces of 18Lin⁻CD34⁺CD38⁻CD133⁺GPI-80⁺ (R6) and 18Lin⁻CD34⁻CD133⁺GPI-80⁺ (R8) cells using seven single-CB units. As shown in Supplementary Fig. 1, most 34⁺38⁻133⁺80⁺ and 34⁻133⁺80⁺ cells expressed CD90, CD49f and CXCR4. Approximately 70% of 34⁺38⁻133⁺80⁺ cells expressed CD93; however, only a few 34⁻133⁺80⁺ cells expressed CD93.

### Megakaryocyte/erythrocyte differentiation potential of HSCs.
The colony-forming capacity (CFC) of single sorted CB-derived 34⁺38⁻133⁺80$^{+/-}$ and 34⁻133⁺80$^{+/-}$ cells was precisely analyzed in methylcellulose cultures (Fig. 1i, j). The plating efficiencies (PEs) of the 34⁺38⁻133⁺80$^{+/-}$ and 34⁻133⁺80$^{+/-}$ cells were 46, 60, 33 and 47%, respectively. The PEs of 34⁺38⁻133⁺80⁻ and 34⁻133⁺80⁻ cells were significantly higher than those of 34⁺38⁻133⁺80⁺ and 34⁻133⁺80⁺ cells. Furthermore, the PEs of 34⁺38⁻133⁺ cells were also significantly higher than those of 34⁻133⁺ cells regardless of GPI-80 expression. Interestingly, the 34⁻133⁺80$^{+/-}$ cells mainly formed CFU-EM colonies. Unexpectedly, 90% of colonies derived from 34⁻133⁺80⁻ cells were CFU-EM (Fig. 1j), whereas the 34⁺38⁻133⁺80$^{+/-}$ cells mainly formed CFU-Mix colonies (73 and 61%). These results indicated that CFCs of human CB-derived CD34⁺ and CD34⁻HSCs are clearly different, and CD34⁻ HSCs had a potent megakaryocyte/erythrocyte differentiation potential in vitro. Representative colonies and their constituent cells are shown in Supplementary Fig. 2.

### Cell cycle status of CD34⁺ and CD34⁻ HSCs.
Most murine adult BM-derived HSCs are known to be cell cycle dormant[26]. However, murine fetal HSCs derived from the AGM (aorta-gonad-mesonephros), placenta and fetal liver are largely cycling[26]. In contrast to extensive studies on murine HSCs, the cell cycle status of human primitive HSCs has not been fully elucidated. Human BM-derived CD34⁺CD38⁻ cells containing primitive LT-HSCs have been reported to be cell cycle dormant[27,28]. However, the cell cycle status of human CB-derived primitive HSCs remains unclear. Therefore, it is necessary to validate the reported findings of murine HSCs in the human HSC hierarchy. In such efforts, it is critical to use a highly purified human HSC population for investigating the cell cycle status/dormancy, as contamination with committed progenitors in the HSC population would hamper the accurate determination of the cell cycle status. As shown in this study, 34⁺38⁻133⁺80⁺ and 34⁻133⁺80⁺ cells, which showed high incidences of CD34⁺ and CD34⁻SRCs (HSCs), provided robust multi-lineage human cell repopulation at 20 weeks after transplantation and efficient engraftment upon the secondary transplantation.

We therefore investigated the proportions of CD34⁺ and CD34⁻SRCs (HSCs) in each cell cycle phase using the abovementioned 34⁺38⁻133⁺80$^{+/-}$ (R6 and R7 in Fig. 1e) and 34⁻133⁺80$^{+/-}$ (R8 and R9 in Fig. 1f) cells via the Ki-67/7-AAD staining method. As shown in Supplementary Fig. 3a, approximately 30 to 40% of the cells in the 34⁺38⁻133⁺80$^{+/-}$ fractions were in G₀ phase of the cell cycle, and <1% of cells in S–G₂–M phase. Approximately 50 to 60% of the cells in the 34⁻133⁺80$^{+/-}$ fractions were in G₀ phase of the cell cycle, and <1% of cells in S–G₂–M phase. In

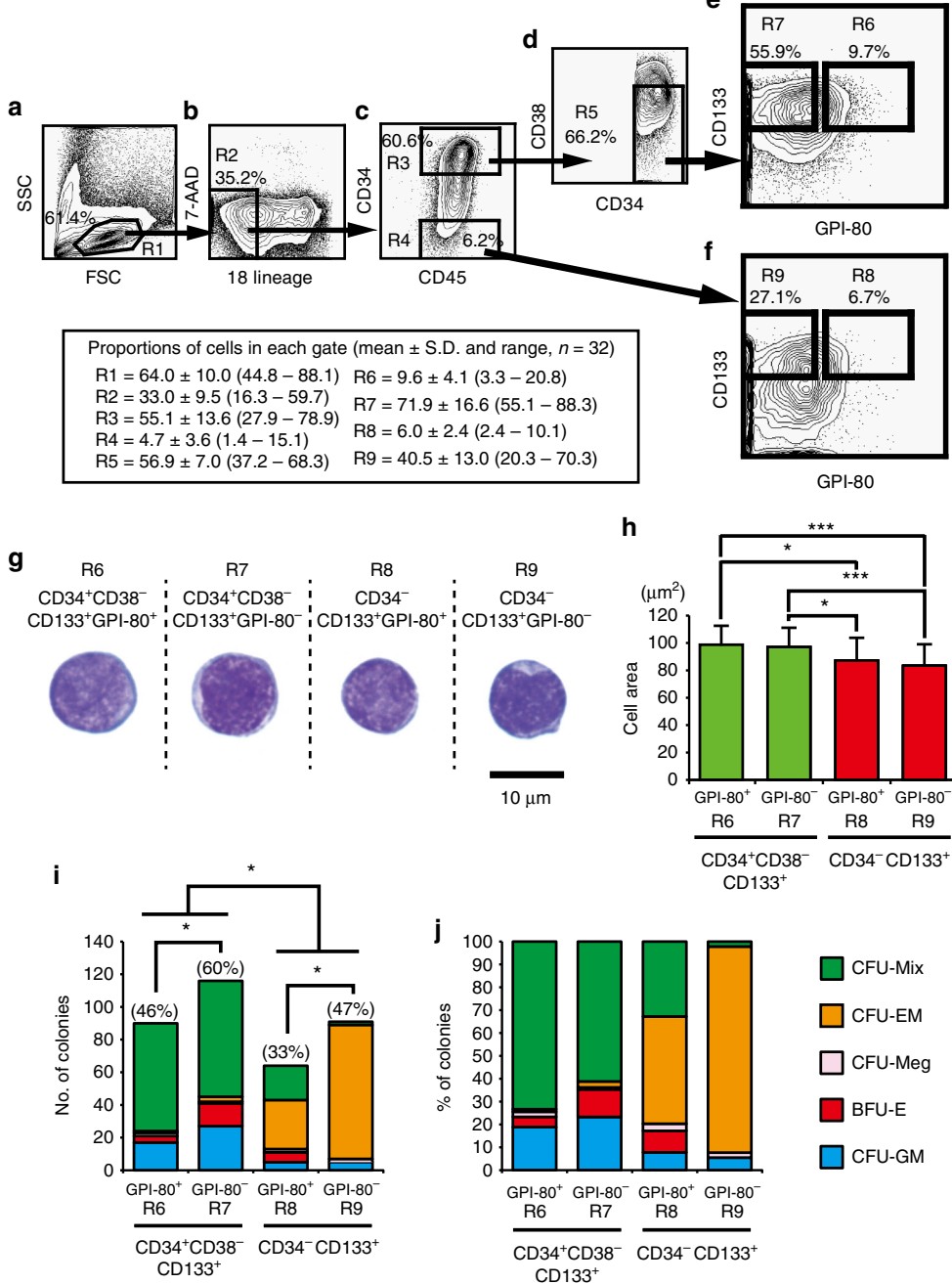

**Fig. 1** Representative FACS profile and colony-forming capacity of highly purified CB-derived 18Lin⁻CD34⁺CD38⁻CD133⁺GPI-80$^{+/-}$ and 18Lin⁻CD34⁻CD133⁺GPI-80$^{+/-}$ cells. A representative FACS profile is shown. **a** The forward scatter/side scatter (FSC/SSC) profile of immunomagnetically separated Lin⁻cells. The R1 gate was set on the blast-lymphocyte window. **b** The R2 gate was set on the 18Lin⁻ living cells. **c** The R2 gated cells were subdivided into two fractions: 18Lin⁻CD45⁺CD34⁺ (R3) and CD34⁻(R4) cells, according to their expression of CD34. The definitions of CD34$^{+/-}$ cells are as follows: the CD34⁺ fraction contains cells expressing >5% of the maximum BV421 fluorescence intensity (FI). The CD34⁻ level of FI was determined based on the Fluorescence Minus One controls. **d** The cells residing in the R3 gate were further subdivided into 18Lin⁻CD45⁺CD34⁺CD38⁻ (R5) cells. The CD38⁻ fraction contains cells expressing <10% of the maximum PE-Cy7 FI. **e** The R5-gated cells were further subdivided into two fractions: 18Lin⁻CD45⁺CD34⁺CD38⁻CD133⁺GPI-80⁺ (R6) and GPI-80⁻(R7) cells according to their expression of CD133 and GPI-80. **f** The R4-gated cells were further subdivided into two fractions: 18Lin⁻CD45⁺CD34⁻CD133⁺GPI-80⁺ (R8) and GPI-80⁻(R9) cells. The definitions of CD133$^{+/-}$ and GPI-80$^{+/-}$ cells are the same as reported[22,25]. The phenotypic purity of the R6 and R8 sorted cells consistently exceeded 99%. Detailed FCM data, including numbers of experiments, and mean ± S.D. and ranges of proportions of cells in each gate, are depicted in the figure. **g** Cells sorted from the R6 to R9 fractions were cytospun on to the slide glasses and stained with May–Grunwald–Giemsa. **h** The areas of the R6 to R9 cells were analyzed using the Image J software program (mean ± S.D., $n = 33$ cells/group, *$p < 0.05$, ***$p < 0.001$, Tukey's multiple comparison procedure). **i** Single-cell CFCs of sorted R6 to R9 cells. The plating efficiency (PE) is shown in parentheses (mean ± S.D., 196 cells/group, *$p < 0.05$, two-way analysis of variance). **j** The percentages of various types of colonies are shown. CFU-Mix (green bar), CFU-EM (orange bar), CFU-Meg (pink bar), BFU-E (red bar) and CFU-GM (blue bar)

contrast, >90% of human BM-derived $34^+38^-133^+80^{+/-}$ cells were in $G_0$ phase of the cell cycle (Supplementary Fig. 3b). As controls, we simultaneously investigated the cell cycle status of CB- and BM-derived $CD34^+CD38^-CD90^+$ HSCs ($CD90^+$ HSCs), $CD34^+CD38^-CD90^-CD45RA^-$ cells (MPPs), $CD34^+CD38^-CD90^-CD45RA^+$ cells (MLPs) and $CD34^+CD38^{high}$ hematopoietic progenitor cells (HPCs) (Supplementary Fig. 3c-e). The purification methods of these target cells are shown in Supplementary Fig. 3c. As shown in Supplementary Figs. 3d and e, the proportions of CB-derived $CD34^+$ HSC/HPCs in the $G_0$ phase was <40%. The proportion of cells in the $G_0$ phase gradually decreased in accordance with the differentiation status (Supplementary Fig. 3d and e). In contrast, the proportion of BM-derived $CD90^+$ HSCs in the $G_0$ phase was >90% (Supplementary Figs. 3d and e). And the proportions of MPPs, MLPs and $CD34^+CD38^{high}$ HPCs in the $G_0$ phase were about 80, 70 and 30%, respectively. The cell cycle status of BM-derived $CD34^+$ HSCs was consistent with the findings of the previous reports[27,28]. However, the cell cycle status of the CB-derived $CD34^+$ and $CD34^-$ HSCs in the present study was different from the previously reported data[29]. The CB-derived $CD34^+$ and $CD34^-$ SRCs (HSCs) seemed to be in a greater cycling flux than previously estimated[29]. This is not very surprising, as CB-derived primitive $CD34^+$ and $CD34^-$ HSCs may reflect fetal hematopoiesis[30–32].

In order to validate the dormancy of CB-derived $CD34^+$ and $CD34^-$SRCs (HSCs), we performed a single-cell proliferation assay. As shown in Supplementary Fig. 4a, single $34^+38^-133^+80^+$ and $34^-133^+80^+$ cells began to divide after 48 h of culture with basal medium containing SCF + TPO. Half of these cells completed their first cell division by 96 h of culture. In contrast, almost all of the $18Lin^-CD34^+CD38^{high}$ cells completed their first cell division by 48 h of culture. In the culture with complete medium containing 10% FCS and 7 cytokines (SCF, TPO, FL, IL-3, IL-6, GM-CSF and G-CSF), almost all of the three classes of HSCs/HPCs completed their first cell division by 96 h of culture (Supplementary Fig. 4b). Among the three classes of HSCs/HPCs, the $CD34^+CD38^{high}$ cells (HPCs) most rapidly proliferated, and the $34^-133^+80^+$ cells ($CD34^-$ HSCs) most slowly proliferated. We then determined how long it took for these three classes of HSCs/HPCs to complete the first cell division (time to complete first cell division, $T_{fcd}$) (Supplementary Fig. 4c). The $T_{fcd}$ of both the $34^+38^-133^+80^+$ and $34^-133^+80^+$ cells were approximately 70 h in the basal medium. This was significantly longer than the $T_{fcd}$ (40 h) of the $18Lin^-CD34^+CD38^{high}$ cells. The $T_{fcd}$ of the two classes of HSCs was significantly shorter in the complete medium than in the basal medium. Interestingly, the time to complete second cell division ($T_{scd}$) of all 3 classes of HSCs/HPCs ranged from 20 to 30 h, regardless of the culture media (Supplementary Fig. 4d). These results suggest that both primitive $CD34^+$ and $CD34^-$ HSCs and committed HPCs begin to divide in the same fashion once they start their proliferation in cultures.

**Combination of CD133 and GPI-80 for purification of HSCs.** We performed a limiting dilution analysis (LDA) to determine the frequencies of $CD34^+$ and $CD34^-$SRCs in the abovementioned four-cell fractions (R6 to R9 in Fig. 1). As shown in Supplementary Fig. 5 and Supplementary Data 1, the frequencies of SRCs in the $34^+38^-133^+80^+$ and $34^-133^+80^+$ cells were 1/4.9 and 1/8.1 cells, respectively, each of which is the highest frequency of CB-derived $CD34^+$ and $CD34^-$ SRCs to date. The frequencies of SRCs in $34^+38^-133^+80^-$ and $34^-133^+80^-$ cells were 1/51.3 and 1/49.8 cells, respectively. Based on these LDA data, we next analyzed the LT-repopulating capacity of the two classes of HSCs residing in the $34^+38^-133^+80^+$ and $34^-133^+80^+$ cells.

**Primary and secondary repopulating capacities of HSCs.** Next, 200 $34^+38^-133^+80^+$ cells (referred to as $CD34^+$ SRCs/HSCs) containing 41 SRCs ($n = 25$) and 200 $34^-133^+80^+$ cells (referred to as $CD34^-$ SRCs/HSCs) containing 25 SRCs ($n = 23$) were transplanted into NOG mice by IBMI (Fig. 2a; Supplementary Data 2a). All of the primary recipient mice showed signs of human cell repopulation at 12 weeks after transplantation. The repopulation levels were maintained until 20 to 22 weeks after transplantation (Fig. 2b). Both the $CD34^+$ and $CD34^-$ SRCs showed multi-lineage reconstitution abilities in the BM, peripheral blood (PB), spleen and thymus (Fig. 2d, e; Supplementary Data 2b).

We then performed secondary transplantation. Most of the secondary recipient NOG mice that received 1/5 of whole BM cells from the primary engrafted recipient mice showed distinct multi-lineage reconstitution (Fig. 2c, Supplementary Data 2a and 2c and Supplementary Fig. 6). Furthermore, in each 10 primary recipient mice that received 200 $34^+38^-133^+80^+$ cells or 200 $34^-133^+80^+$ cells, we resorted $18Lin^-CD34^+$ and $18Lin^-CD34^-$ cells from the primarily engrafted mouse BM and transplanted these cells into the secondary ones. As shown in Supplementary Data 2a, in the case of $CD34^+$ SRCs, 6 out of 10 mice that received resorted $18Lin^-CD34^+$ cells showed multi-lineage human cell repopulation. However, none of the 10 mice that received $18Lin^-CD34^-$ cells were repopulated with human cells. These results clearly indicated that $CD34^+$ SRCs were able to maintain (self-renew) $CD34^+$ SRCs but could not generate $CD34^-$ SRCs in vivo, as we reported previously[18]. In contrast, in the case of $CD34^-$ SRCs, 6 out of 10 mice that received resorted $18Lin^-CD34^+$ cells showed multi-lineage human cell repopulation. These results clearly indicated that $CD34^-$ SRCs were able to generate $CD34^+$ SRCs in vivo, as we reported previously[18,33]. Very interestingly, 4 out of 10 mice that received $18Lin^-CD34^-$ cells resorted from primary recipient mice that received $34^-133^+80^+$ cells showed multi-lineage human cell reconstitution (Supplementary Fig. 7). These observations suggest that $CD34^-$ SRCs were able to self-renew in NOG mice.

**Lineage differentiation potential of HSCs in vivo.** Next, we precisely analyzed the multi-lineage differentiation potentials of $CD34^+$ and $CD34^-$ SRCs in various organs, including the BM, PB, spleen and thymus, in the abovementioned primary recipient NOG mice that received 200 $34^+38^-133^+80^+$ or $34^-133^+80^+$ cells (Supplementary Data 2b). As shown in Supplementary Fig. 8a, the human $CD45^+$ cell repopulation capacities of both $CD34^+$ and $CD34^-$ SRCs were not statistically different. These data were consistent with our previously reported data[18,21,22,25,33]. However, the differentiation potentials of these $CD34^+$ and $CD34^-$ SRCs with regard to $CD19^+$ lymphoid cells were clearly different. These $CD34^+$ SRCs showed repopulation of $CD19^+$ cells at significantly higher levels than $CD34^-$ SRCs in the left tibia (injected site) and PB (Supplementary Fig. 8a(iv)), which is consistent with our previous report[34]. Very interestingly, these $CD34^-$ SRCs showed significantly higher levels of $CD41^+$ cells than $CD34^+$ SRCs in the left tibia (injected site) and spleen (Supplementary Fig. 8a(vi)). Furthermore, these $CD34^-$ SRCs also showed significantly higher levels of $CD235a^+$ cells than $CD34^+$ SRCs in the left tibia (injected site) and other bones (Supplementary Fig. 8a(vii)). These results demonstrated that $CD34^-$ SRCs (HSCs) possess more potent megakaryocyte/erythrocyte differentiation potential in vivo than $CD34^+$ SRCs (HSCs).

We then precisely analyzed the multi-lineage differentiation potentials of $CD34^+$ and $CD34^-$ SRCs in secondary transplantations (Supplementary Data 2c). As shown in Supplementary

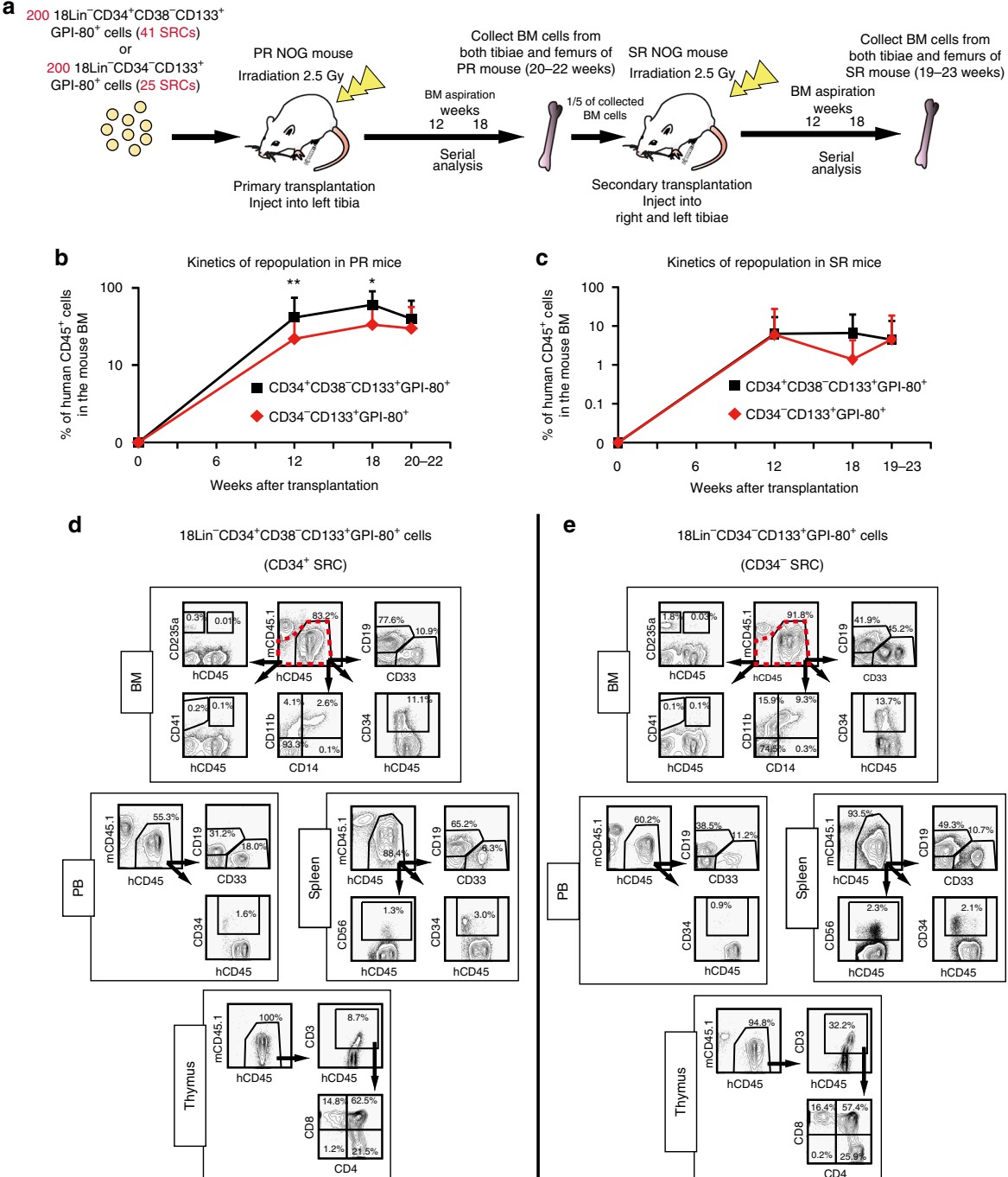

**Fig. 2** SCID-repopulating cell activity and serial analyses of human cell repopulation of 18Lin⁻CD34⁺CD38⁻CD133⁺GPI-80⁺ and 18Lin⁻CD34⁻CD133⁺GPI-80⁺ cells in primary and secondary NOG mice by IBMI. **a** A schematic illustration of primary and secondary transplantation of CD34⁺ and CD34⁻ SRCs is shown. PR, primary recipient; SR, secondary recipient. **b** The long-term repopulating potential of the CD34⁺ and CD34⁻ SRCs was determined by serially analyzing the kinetics of BM engraftment for 20 to 22 weeks in primary NOG mice that received transplants of 200 18Lin⁻CD34⁺CD38⁻CD133⁺GPI-80⁺ cells (41 SRCs) ($n = 25$) and 200 18Lin⁻CD34⁻CD133⁺GPI-80⁺ cells (25 SRCs) ($n = 23$), respectively. All of the primary recipient mice were highly repopulated with human CD45⁺ cells. The human CD45⁺ cell repopulation rates of the mice that were transplanted with CD34⁺ SRCs at 12 and 18 weeks after transplantation were significantly higher than those of the mice that were transplanted with CD34⁻ SRCs. However, the repopulation rates of both mice were comparable at 20 to 22 weeks after transplantation (mean ± S.D., *$p < 0.05$, **$p < 0.01$, two-tailed Student's $t$-test). **c** The kinetics of human CD45⁺ cell reconstitution in secondary recipient NOG mice that received 1/5 of whole BM cells from the abovementioned primary recipient mice. The repopulation rates of both mice ($n = 17$ for CD34⁺ SRCs and $n = 15$ for CD34⁻ SRCs) were comparable during the observation period (12 to 23 weeks after transplantation) (there were no significant differences between both group, two-tailed Student's $t$-test). **d**, **e** Both of the CD34⁺ and CD34⁻ SRCs showed multi-lineage reconstitution abilities in primary recipient mice, including CD34⁺ stem/progenitor cells, CD19⁺ B-lymphoid, CD33⁺ and CD11b⁺ myeloid, CD14⁺ monocytic, CD235a⁺ erythroid and CD41⁺ megakaryocytic cells. The presence of CD235a⁺ and CD41⁺ cells was analyzed using human CD45⁺/⁻ cell gate (depicted by red dotted lines). We also detected CD56⁺ NK cells in the spleen and CD3⁺ and CD4/CD8 single/double-positive cells in the thymus. Most of the engrafted primary recipient mice showed distinct multi-lineage secondary reconstitution abilities. Detailed FCM data, including numbers of experiments, means and ranges of proportions of cells in each cell lineage are shown in Supplementary Fig. 8a, Supplementary Data 2a and b

Fig. 8b, the human CD45$^+$ cell repopulation capacities of CD34$^+$ and CD34$^-$ SRCs were not statistically different. These CD34$^-$ SRCs, however, showed significantly higher levels of CD34$^+$ cells than CD34$^+$ SRCs in the both tibiae (injected site) and other bone (Supplementary Fig. 8b(ii)). These results suggest that CD34$^-$ SRCs are a more immature class of HSCs than CD34$^+$ SRCs.

**Repopulating capacities of single CD34$^+$ and CD34$^-$HSCs.** To validate the definitive stem cell nature, such as self-renewal activity and multi-lineage differentiation capacity of human primitive CD34$^+$ and CD34$^-$ SRCs (HSCs), we performed a single-cell transplantation analysis with highly purified 34$^+$38$^-$133$^+$80$^+$ and 34$^-$133$^+$80$^+$ cells. In the first set of experiments (Supplementary Data 3a), 3 out of 57 (5.3%) or 2 out of 78 (2.6%) recipient NOG mice that received single 34$^+$38$^-$133$^+$80$^+$ or 34$^-$133$^+$80$^+$ cells showed multi-lineage human cell repopulation. As shown in Supplementary Data 3a and b, 7 out of 62 (11.3%) and 6 out of 100 (6.0%) recipient NSG mice that received single 34$^+$38$^-$133$^+$80$^+$ or 34$^-$133$^+$80$^+$ cells showed multi-lineage human cell repopulation, with repopulation rates of 0.8%–45.1% or 0.9%–30.1% in the injected left tibiae, respectively. Notably, even single-cell purified human CB-derived CD34$^-$ SRCs were able to be engrafted in recipient NOG/NSG mice; to our knowledge, this is the first time this has been reported. We then precisely analyzed their multi-lineage reconstitution abilities in the BM, PB and spleen, as shown in Fig. 3 and Supplementary

Data 3b. These results confirmed that both of the CD34$^+$ and CD34$^-$SRCs had multi-lineage differentiation capability. In addition, engraftment of single 34$^+$38$^-$133$^+$80$^+$ and 34$^-$133$^+$80$^+$ cells provided clear evidence that human CB-derived primitive CD34$^+$ and CD34$^-$SRCs (HSCs) expressed CD133 and GPI-80.

**Evidence for the in vivo self-renewing capacities of HSCs.** We next performed secondary transplantation with whole BM cells from engrafted primary recipient mice that received single purified CD34$^+$ and CD34$^-$ SRCs. Five out of nine mice receiving CD34$^+$ SRCs and five out of eight mice receiving CD34$^-$ SRCs showed distinct secondary multi-lineage human cell repopulations (Supplementary Data 3a). Of note, one NSG mouse (34$^+$ NSG046) that received a single 34$^+$38$^-$133$^+$80$^+$ cell showed robust human CD45$^+$ cell repopulation even at 22 weeks after transplantation (Supplementary Data 3a). We transplanted 2/5 whole BM cells into two secondary recipient NSG mice. Both mice showed lympho-myeloid reconstitution by 12 to 22 weeks after transplantation (Supplementary Fig. 9a and b, and Supplementary Data 3c). The patterns of multi-lineage repopulation observed in the two secondary recipient mice were not equivalent. Notably, four out of six NSG mice that received a single 34$^-$133$^+$80$^+$ cell showed robust human CD45$^+$ cell repopulation at 20 to 24 weeks after transplantation (Supplementary Data 3a). We then transplanted 2/5 whole BM cells of each engrafted primary recipient mice into two secondary recipient NSG mice. Surprisingly,

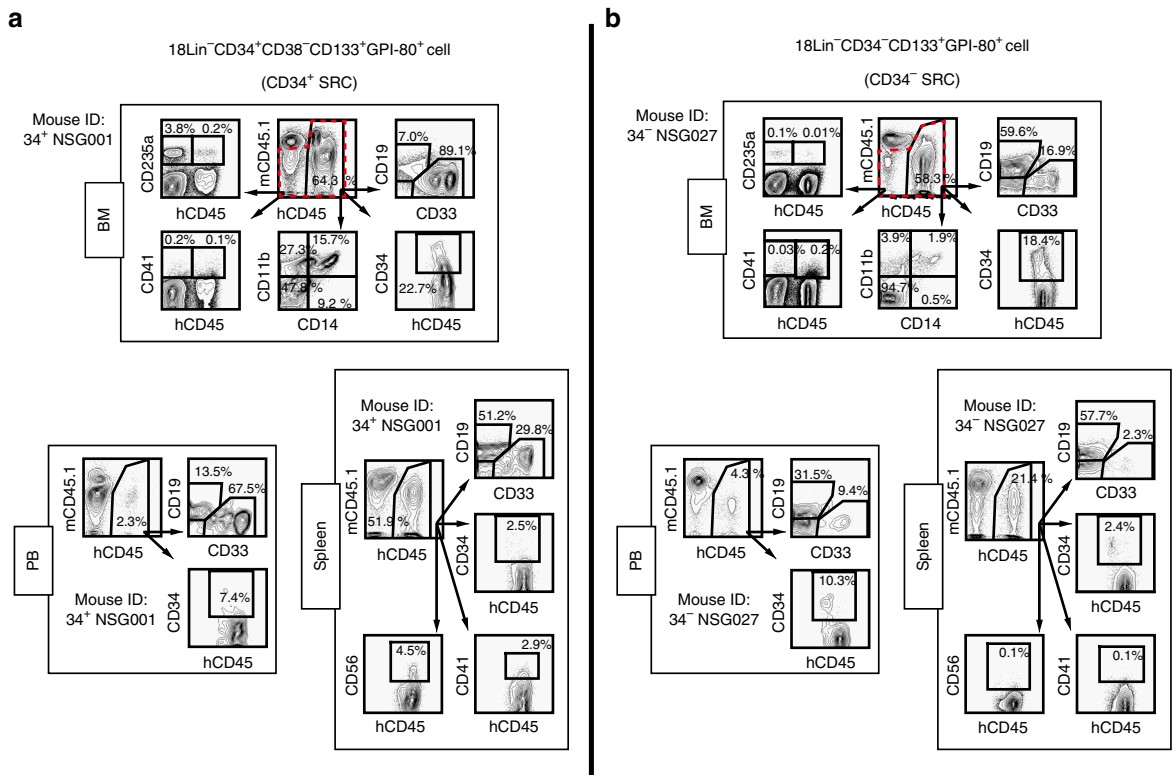

**Fig. 3** Human multi-lineage hematopoietic repopulation abilities of single CD34$^+$ and CD34$^-$ SRCs. Human multi-lineage hematopoietic repopulations in the primary recipient NSG mice that received **a** single 18Lin$^-$CD34$^+$CD38$^-$CD133$^+$GPI-80$^+$ (Mouse ID: 34$^+$ NSG001) or **b** single 18Lin$^-$CD34$^-$CD133$^+$GPI-80$^+$ (Mouse ID: 34$^-$ NSG027) cells were analyzed by 6-color FCM at 22 or 21 weeks after transplantation. The expression of CD19, CD33 CD34 (BM, PB and spleen), CD11b and CD14 (BM), CD41 (BM and spleen), CD235a (BM) and CD56 (spleen) in living human CD45$^+$ cells was analyzed. The presence of CD41$^+$ and CD235a$^+$ cells in BM was analyzed using a human CD45$^{+/-}$ cell gate (depicted by red dotted lines). The mouse ID numbers presented in the figure corresponded those listed in Supplementary Data 3a. Analyses of the BM (other bones), PB and spleens of the two representative mice that received either CD34$^+$ SRC **a** or CD34$^-$ SRC **b** revealed that both SRCs had an in vivo differentiation capacity comparable to that of CD34$^+$ stem/progenitor cells, CD19$^+$ B-lymphoids, CD33$^+$/CD11b$^+$ myeloids, CD14$^+$ monocytes, CD235a$^+$ erythroid and CD41$^+$ megakaryocytic lineages. CD56$^+$ NK cells were detected in the spleen. These results confirmed that both CD34$^+$ and CD34$^-$ SRCs had definite multi-lineage differentiation potential. Detailed FCM data of all recipient mice that received single CD34$^+$ and CD34$^-$ SRCs are presented in Supplementary Data 3b

all four single CD34− SRC-engrafted mice (CD34− NSG 027, 058, 076 and 079) repopulated each of the two secondary recipient NSG mice (eight mice in total) (Supplementary Data 3a). All eight secondary recipient NSG mice showed distinct multilineage human cell repopulation by 12 to 22 weeks after transplantation (Supplementary Data 3c). The repopulation patterns of one representative NSG mouse (34− NSG058) are shown in Supplementary Fig. 9a and c. It was quite interesting that the patterns of the multi-lineage repopulation observed in the two secondary recipient mice were not equivalent, suggesting that daughter SRCs generated from the single parental CD34+ and CD34− SRCs were functionally heterogeneous.

Finally, we performed tertiary transplantation with whole BM cells from secondarily engrafted recipient mice that originally received single purified CD34+ and CD34− SRCs (Supplementary Data 3a). Two secondary recipient NSG mice (34+ NSG046a and NSG046b) that originally received a single 34+38−133+80+ cell in primary transplantation engrafted 2/4 tertiary recipient mice 22 weeks after transplantation (Supplementary Fig. 9d). Of note, two secondary recipient NSG mice (34− NSG058a and NSG058b) that originally received a single 34−133+80+ cell in primary transplantation did engraft 4/4 tertiary recipient mice (Supplementary Fig. 9d). All of the tertiary engrafted mice showed multi-lineage human CD45+ cell repopulation at 22 weeks after transplantation (Supplementary Fig. 9d and Supplementary Data 3d). These results clearly demonstrated that human CB-derived single purified CD34+ and CD34− SRCs can extensively self-renew in xenografted mice.

**CD34− HSCs reside at the apex of the human HSC hierarchy**. As we reported previously, CD34−SRCs (HSCs) are able to generate CD34+ SRCs (HSCs) in vitro[25,35,36] and in vivo.[18,33] In this study, we showed that CD34− SRCs (HSCs) (18Lin−CD34−CD133+GPI-80+ cells) were able to generate CD34+ as well as CD34− SRCs (HSCs) in vivo (Supplementary Fig. 7e and f). However, CD34+ SRCs (HSCs) (18Lin−CD34+CD38−CD133+GPI-80+ cells) were only able to generate CD34+ SRCs (HSCs) in vivo (Supplementary Figs. 7b and c). They were unable to generate CD34− SRCs (HSCs) in vivo. To elucidate the hierarchical position of the CD34−SRCs (HSCs) (18Lin−CD34−CD133+GPI-80+ cells) in comparison to previously well-established CD34+ SRCs (HSCs) (9Lin−CD34+CD38−CD45RA−CD90+CD49f+ cells)[3], we next co-cultured CB-derived 18Lin−CD34−CD133+GPI-80+ and 9Lin−CD34+CD38−CD45RA−CD90+CD49f+ cells with human BM-derived CD271+SSEA-4+ MSCs (DP MSCs)[35]. The results are shown in Supplementary Fig. 10 and Supplementary Data 4. The 18Lin−CD34−CD133+GPI-80+ cells were able to generate 9Lin−CD34+CD38−CD45RA−CD90+CD49f+/− cells with SRC activities (Supplementary Fig. 10i and j). Furthermore, they were able to generate CD34− SRCs (HSCs) in vitro (Supplementary Fig. 10k). These results are consistent with the findings of the in vivo study, as shown in Supplementary Fig. 7. In contrast, the 9Lin−CD34+CD38−CD45RA−CD90+CD49f+ cells were able to generate CD34+CD38−CD45RA−CD90+CD49f+/− cells with SRC activities (Supplementary Fig. 10d and e). However, they were unable to generate CD34− SRCs (HSCs) in vitro (Supplementary Fig. 10f). These results suggest that CD34− SRCs (HSCs) (18Lin−CD34−CD133+GPI-80+ cells) are more immature than CD34+ SRCs (HSCs) (9Lin−CD34+CD38−CD45RA−CD90+CD49f+ cells) in the human HSC hierarchy, indicating that CD34− SRCs (HSCs) reside at the apex of the human HSC hierarchy.

**A unique molecular signature of CD34− HSCs**. To provide an independent line of evidence for characterizing our highly purified CD34+ and CD34− SRCs (HSCs), we analyzed the gene expression profiles of these two classes of SRCs (HSCs) and other types of progenitors and mature cells at the single-cell level. As shown in Fig. 4a, a principal component analysis (PCA) demonstrated that the gene expression profiles of individual CD34− HSCs were clearly different from those of CD34+ and CD90+ HSCs[3]. In contrast, the gene expression profiles of individual CD34+ and CD90+ HSCs overlapped. An unsupervised hierarchical clustering analysis (Fig. 4b) clearly showed two clusters; 1 and 2. Interestingly, all CD34− HSCs belonged to cluster 1. Conversely, cluster 2 contained some of CD34+ HSCs, CD90+ HSCs and various HPCs, including multipotent progenitors (MPPs), common myeloid progenitors (CMPs), granulocyte/macrophage progenitors (GMPs), megakaryocyte/erythrocyte progenitors (MEPs) and multilymphoid progenitors (MLPs). These results demonstrated that the gene expression profiles of CD34− HSC were unique and largely different from those of other classes of CD34+ HSCs. We then performed a heatmap analysis, as shown in Fig. 4c. We identified three independent clusters that respectively contained HSC/HPC-related (A), HPC-related (B) and mature cell-related (C) genes.

We next focused on the gene expression profiles of primitive CD34+ and CD34−and CD90+ HSCs in the heatmap (Fig. 5a). These three classes of primitive HSCs showed different gene expression profiles. As shown in the upper column of violin plots (Fig. 5b), HSC maintenance genes such as the *KIT*, *RUNX1*, *TAL1*, *BMI1*, *DNMT3A* and *TGFBR1* and *R2* were comparably expressed in these three classes of HSCs. In contrast, genes such as the *IFITM1*, *MPL*, *IKZF1*, *ETV6*, *ALDH1A1* and *IGF1R* were more highly expressed in CD34+ and/or CD90+ HSCs than in CD34− HSCs (Fig. 5b). Very characteristically, the *EZH2* and *MYB* genes were turned out to be highly expressed in CD34−HSCs (Fig. 5b). Overall, the PCA and unsupervised hierarchical clustering analyses demonstrated that the gene expression profiles of individual CD34+ and CD34− HSCs were clearly different.

**Global gene expression analyses of CD34+ and CD34− HSCs**. Finally, we performed global gene expression analyses of highly purified human CB-derived 18Lin−CD34+CD38−CD133+GPI-80+ cells (referred to as CD34+ HSCs) and 18Lin−CD34−CD133+GPI-80+ cells (referred to as CD34− HSCs). As a control, we sorted 18Lin−CD34+CD133− cells (referred to as non-SRCs [non-HSCs]), that had an undetectable level of SRC activity, as previously reported[22].

First, we compared the gene expression profiles of CD34+ and CD34− HSCs using a gene set enrichment analysis (GSEA). Gene sets enriched in CD34+ HSCs are those related to interferon α/β/γ response/signaling, cell adhesion/migration, cytokine signaling in the immune and inflammatory responses (Fig. 6a right). Gene sets enriched in CD34− HSCs are those related to hypoxia, ontogeny, megakaryocyte/erythrocyte differentiation, Wnt signaling and angiogenesis (Fig. 6a left). As described above, the CD34− HSCs showed potent megakaryocyte/erythrocyte differentiation potential in vitro and in vivo. It is interesting that the gene set related to megakaryocyte/erythrocyte differentiation was enriched in CD34− HSCs.

We then focused on the signaling pathways important for the HSC function and maintenance. Interestingly, CD34− HSCs expressed higher levels of the gene set related to Wnt signaling (Fig. 6a, b), while CD34+ HSCs expressed higher level of the gene sets related to IFN-α, IFN-γ, TGF-β and Notch signaling (Fig. 6a, c). As mentioned above, a single-cell gene expression analysis clearly showed that CD34−HSCs expressed lower levels of *IFITM1* than CD34+ HSCs (Fig. 5b). Therefore, we next analyzed the IFN-related signaling genes. As shown in Supplementary

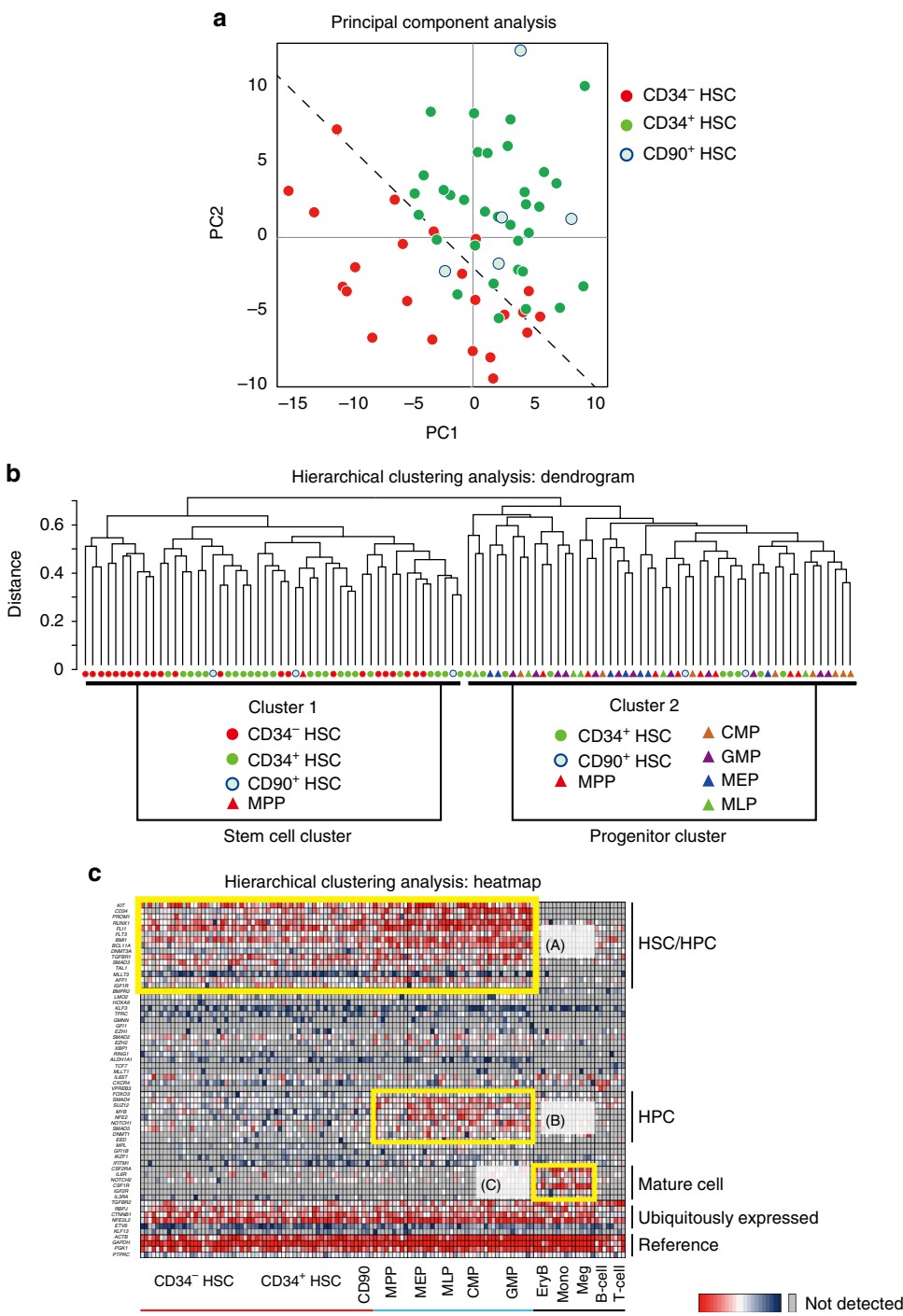

**Fig. 4** Single-cell-based gene expression profiles of human CB-derived CD34$^+$ and CD34$^-$ HSCs. Using highly purified human CB-derived 18Lin$^-$CD34$^+$CD38$^-$CD133$^+$GPI-80$^+$ and 18Lin$^-$CD34$^-$CD133$^+$GPI-80$^+$ cells, we performed a single-cell-based gene expression analysis. The immunophenotypes of target cells, including controls, are presented in Supplementary Data 6 and the 79 target genes that play important roles in the pathway of HSC development/differentiation[63] are listed in Supplementary Data 7. **a** A principal component analysis (PCA) revealed that the gene expression profiles in individual CD34$^+$ ($n = 33$) and CD34$^-$ HSCs ($n = 23$) were clearly different. The gene expression profiles of individual CD90$^+$ HSCs ($n = 5$) were similar to those of CD34$^+$ HSCs but not to those of CD34$^-$HSCs. The dotted line represents the border region between the CD34$^+$ and CD34$^-$ HSCs calculated by a Fisher's linear discriminant analysis. **b** An unsupervised hierarchical clustering analysis (Dendrogram) clearly showed two clusters, 1 and 2. Interestingly, all CD34$^-$HSCs belong to cluster 1. Three subgroups were detected in cluster 1. The left-most subgroup uniformly contained 10 CD34$^-$ HSCs. The remaining 13 CD34$^-$ HSCs were scattered between the other two subgroups mixed with CD34$^+$ HSCs and CD90$^+$ HSCs and MPPs. In contrast, some of the CD34$^+$ HSCs and CD90$^+$ HSCs, most of the MPP, and all other HPCs (CMP, GMP, MEP and MLP) belonged to cluster 2. These results demonstrated that the gene expression profiles of CD34$^-$ HSC were unique and largely differed from those of other classes of CD34$^+$ HSPCs. **c** A hierarchical clustering analysis of 62 genes (heatmap) detected 3 clusters (**a–c**), which are highlighted with yellow squares. Cluster (A) contained HSC/HPC-related genes, including *RUNX1*, *BMI1*, *DNMT3a* and *SCL/TAL1*, in addition to *CD34* and *PROM1* (CD133). Cluster (B) contained HPC-related genes, including *MYB*, *FOXO3A*, *SMAD4*, *NFE2* and *NOTCH1*. Cluster (C) contained mature cell-related genes, including those for various cytokine receptors (*CSF2RA*, *IL-6R*, *CSF1R*, *IGF2R* and *IL-3RA*)

Fig. 11, CD34$^+$ and CD34$^-$ HSCs expressed comparable levels of *IFNAR1* and *R2* receptors, *TYK2* and *JAK1*. However, CD34$^-$ HSCs expressed significantly lower levels of their downstream target genes, including *STAT1*, *IFITM1*, *IFITM3*, *DDX58*, *IFI44*, *CXCL10* and *CXCL11* than CD34$^+$ HSCs. As recently reported, high levels of IFN-α activated dormant HSCs in vivo[37,38]. These results therefore suggest that CD34$^-$ HSCs may be more resistant than CD34$^+$ HSCs to chronic activation of the IFN-α pathway.

Next, we compared the gene expression profiles of CD34$^+$ and CD34$^-$ SRCs (HSCs) with those of non-SRCs (non-HSCs) by a GSEA. The results showed that gene sets related to quiescence, inflammatory signaling, HSC markers, cell adhesion, cytokine signaling and chemokine/migration were enriched in CD34$^+$ and CD34$^-$ SRCs (HSCs) (Supplementary Fig. 12a right). In contrast, gene sets related to cell cycle progression, differentiation, DNA replication, metabolism and organellar activity were enriched in

**a**  Gene expression profiles of CD34$^-$, CD34$^+$ and CD90$^+$ HSCs (heatmap)

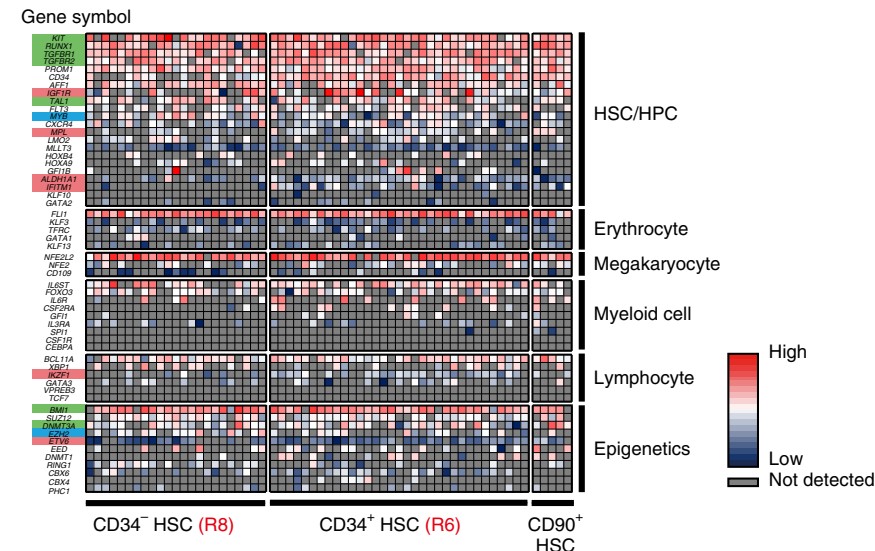

**b**  Gene expression profiles of CD34$^-$, CD34$^+$ and CD90$^+$ HSCs (violin plots)

HSC maintenance genes highly expressed in CD34$^-$, CD34$^+$ and CD90$^+$ HSCs (highlighted by green)

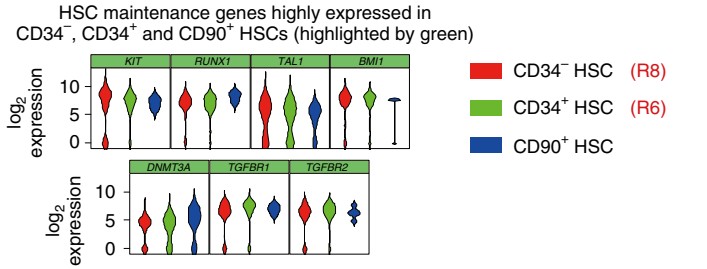

Genes highly expressed in CD34$^+$ HSCs compared with CD34$^-$ HSCs (highlighted by pink)

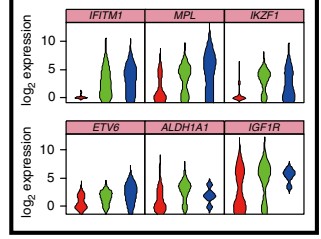

Genes highly expressed in CD34$^-$ HSCs compared with CD34$^+$ HSCs (highlighted by blue)

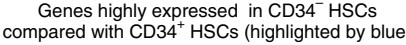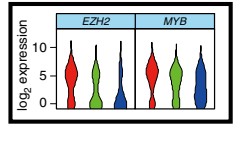

**c**  Expression of reference genes and genes for CD34, CD133 and GPI-80

Reference genes

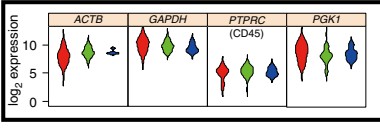

CD34 and CD133

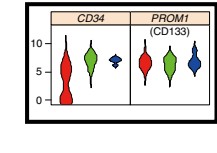

GPI-80

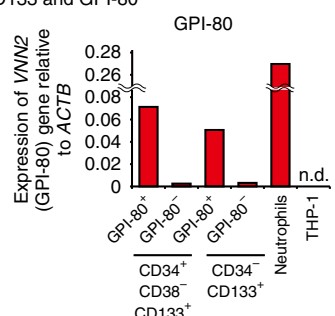

non-SRCs (non-HSCs) (Supplementary Fig. 12a left). We then selected the 1323 differentially expressed probes with a >twofold difference in expression between CD34$^+$ and CD34$^-$SRCs and non-SRCs (Supplementary Fig. 12b). A heatmap analysis identified several marker molecules previously reported for characterizing/purifying human HSCs, including *ITGA6*, *PROCR*, endothelial cell-selected adhesion molecule (*ESAM*) and *PROM1* (CD133). CD34$^+$ and CD34$^-$ SRCs (HSCs) expressed higher levels of *ITGA6* (CD49f) than non-SRCs, which was consistent with a recent report that CD49f was a specific HSC marker for human CB-derived CD90$^+$ HSCs[3]. Interestingly, CD34$^+$ and CD34$^-$SRCs (HSCs) expressed higher levels of *PROCR* (EPCR) than non-SRCs. As recently reported[39], EPCR/PAR1 signaling facilitated LT-HSC BM repopulation, retention and the survival. Very recently, Yokota et al. reported that *ESAM* was a useful purification marker for human BM and CB-derived HSCs[40]. As expected from our ultra-high-resolution purification method, CD34$^+$ and CD34$^-$ SRCs (HSCs) expressed higher levels of *PROM1* (CD133) than non-SRCs. In contrast, non-SRCs expressed higher levels of the *CD93* gene, which was reported to be a positive marker for human CB-derived CD34$^-$ HSCs[41]. However, as shown in Supplementary Fig. 1, our highly purified 34$^-$133$^+$80$^+$ cells did not express CD93 at all. Collectively, these GSEAs demonstrated a clear difference between these two classes of CD34$^+$ and CD34$^-$SRCs (HSCs), suggesting that CD34$^-$ HSCs are a distinct class of human primitive HSCs in comparison to CD34$^+$ HSCs.

**Ex vivo expansion of CD34$^-$ HSCs.** Thus far, approximately 40,000 CB transplantations (CBTs) have been performed worldwide for children and adults with severe hematological diseases.[42–44] Thus, CB is now one of the most commonly used sources for allogeneic hematopoietic stem cell transplantation (HSCT).[42–44] However, the number of HSCs residing in CB units is limited resulting in delayed neutrophil and platelet recovery and engraftment failure. Both are life-threatening complications. To overcome these clinical challenges, efficient ex vivo expansion of CB-derived HSCs is awaited.

The purine derivative StemRegenin-1 (SR-1) was first identified to promote the expansion of CD34$^+$ HSCs/HPCs.[45] Cultures of CB-derived CD34$^+$ cells in the presence of SCF + TPO + FL + SR-1 led to a 50-fold increase in CD34$^+$ cells and a 17-fold increase in SRCs. Recently, a pyrimidoindole derivative (UM171) was found to attenuate cell differentiation and promoted the ex vivo expansion of CB-derived CD34$^+$CD45RA$^-$ cells.[46] Interestingly, the frequencies of LT-HSCs were 13-fold higher in cultures supplemented with UM171 than in those supplemented with DMSO control. Given these findings, we tried to expand very primitive CB-derived CD34$^-$ SRCs (HSCs) (18Lin$^-$CD34$^-$CD133$^+$GPI-80$^+$ cells) using SCF + TPO + FL + SR-1 or UM171. As shown in Supplementary Fig. 13c and d, 18Lin$^-$CD34$^-$CD133$^+$GPI-80$^+$ cells actively proliferated and showed 460-fold (UM171) to 950-fold (SR-1) increases in total numbers

of cells, yielding $9 \times 10^5$ (UM171) to $13 \times 10^5$ (SR-1) 12Lin$^-$CD45RA$^-$CD34$^+$ cells. We then performed LDAs to analyze their effects on the expansion of SRCs (Supplementary Fig. 13e and f). Unfortunately, we were unable to expand CD34$^-$ SRCs (HSCs). As shown in this study (Supplementary Figs. 7 and 10), CD34$^-$ SRCs (HSCs) did generate CD34$^+$ SRCs (HSCs) in these cultures in the presence of cytokines plus SR-1 or UM171 (Supplementary Fig. 13a and b). However, the expansion efficiencies were 1.33 for SR-1 and 0.42 for UM171 (Supplementary Fig. 13a and f). These efficiencies were not statistically significant in comparison to DMSO (control). These results showed that the ex vivo expansion of primitive CB-derived CD34$^-$ SRCs (HSCs) was more difficult than previously considered, and suggesting that ex vivo expansion of primitive CB-derived CD34$^-$ SRCs (HSCs) may require niche cells/factors for maintenance and/or proliferation. Further studies will be needed to clarify the details of this important issue.

## Discussion

We developed an ultra-high-resolution purification method using two positive markers, CD133 and GPI-80, and succeeded in highly purifying human CB-derived CD34$^+$ and CD34$^-$SRCs (HSCs) at 1/5 and 1/8 cell levels, respectively. GPI-80 has recently been reported as a positive marker for human fetal liver hematopoietic stem/progenitor cells (HSPCs)[47]. Because human placenta/CB-derived HSPCs reflect fetal hematopoiesis[30,31], it is conceivable that GPI-80 is expressed on CB-derived primitive HSCs. Interestingly, both CD133 and GPI-80 were correlated with the polarization and migration of leukocytes and HSPCs[23,24,48,49]. Thus, these two molecules seem to be functional markers. Our updated method enables us to perform single-cell-based transplantation analyses. Both groups of engrafted mice that received single 34$^+$38$^-$133$^+$80$^+$ or 34$^-$133$^+$80$^+$ cells showed distinct secondary repopulation capability with multilineage differentiation. It is noteworthy that 1/6 and 4/6 primarily engrafted NSG mice that received single 34$^+$38$^-$133$^+$80$^+$ and 34$^-$133$^+$80$^+$ cells robustly repopulated two secondary recipient mice, respectively (Supplementary Data 3a and Supplementary Fig. 9a to c). Furthermore, both CD34$^+$ and CD34$^-$SRCs showed substantial single-cell-initiated tertiary repopulating capacities (Supplementary Data 3a and Supplementary Fig. 9d). This indicates that both SRCs sustained human multi-lineage hematopoiesis in NSG mice for over one year (Supplementary Data 3a). These results showed that individual CD34$^+$ and CD34$^-$SRCs (HSCs) have potent proliferative and extensively self-renewing capacities.

In this study, the single-cell transplantation efficiencies of highly purified 34$^+$38$^-$133$^+$80$^+$ and 34$^-$133$^+$80$^+$ cells were 11 and 6%, respectively (Supplementary Data 3a), which were still relatively low. The xenotransplantation assay with NOG/NSG mice for detecting SRC activity, however, is not an ideal assay system, partly because of its species specificity[50]. As we showed in the present study, single CD34$^+$ and CD34$^-$SRCs exerted robust

**Fig. 5** A comparison of the gene expression profiles of single-purified human CB-derived CD34$^+$ and CD34$^-$ and CD90$^+$ HSCs by qRT-PCR. Multi-plex (79 genes) single-cell qRT-PCR using CD34$^+$ and CD34$^-$ HSCs and CD90$^+$ HSCs was performed. **a** The 54 gene expression profiles in CD34$^+$ and CD34$^-$HSCs and CD90$^+$ HSC are depicted using a heatmap. These 54 genes were classified as HSC/HPC-, erythrocyte-, megakaryocyte-, myeloid cell-, lymphocyte- and epigenetics-related genes. **b** The expression of individual genes in CD34$^+$ and CD34$^-$ HSCs and CD90$^+$ HSCs is depicted by violin plots. The HSC maintenance genes highly expressed in both CD34$^+$ and CD34$^-$ HSCs and CD90$^+$ HSCs (*KIT*, *RUNK1*, *TAL1*, *BMI1*, *DNMT3A*, *TGFBR1* and *R2*) are shown in the upper panel and highlighted by green color in **a**. The genes highly expressed in CD34$^+$ HSCs (*IFITM1*, *MPL*, *IKZF1*, *ETV6*, *ALDH1A1* and *IGF1R*) are shown in the middle left panel and highlighted by pink color in **a**. The genes highly expressed in CD34$^-$ HSCs (*EZH2* and *MYB*) are shown in the middle right panel and highlighted by blue color in **a**. The gate names (R6 and R8) presented in this figure correspond to the same fractions in Fig. 1e, f. **c** Violin plots of the reference genes, including *ACTB*, *GAPDH*, *PTPRC* (CD45) and *PGK-1* (left). Violin plots of the genes for *CD34* and *PROM1* (CD133) (middle). qRT-PCR of the *VNN2* (GPI-80) mRNA expression in the 18Lin$^-$CD34$^+$CD38$^-$CD133$^+$GPI-80$^{+/-}$ and 18Lin$^-$CD34$^-$CD133$^+$GPI-80$^{+/-}$ cells, compared with the positive control (CB-derived neutrophils) and negative control (THP-1 cells) (right). n.d.: not detected

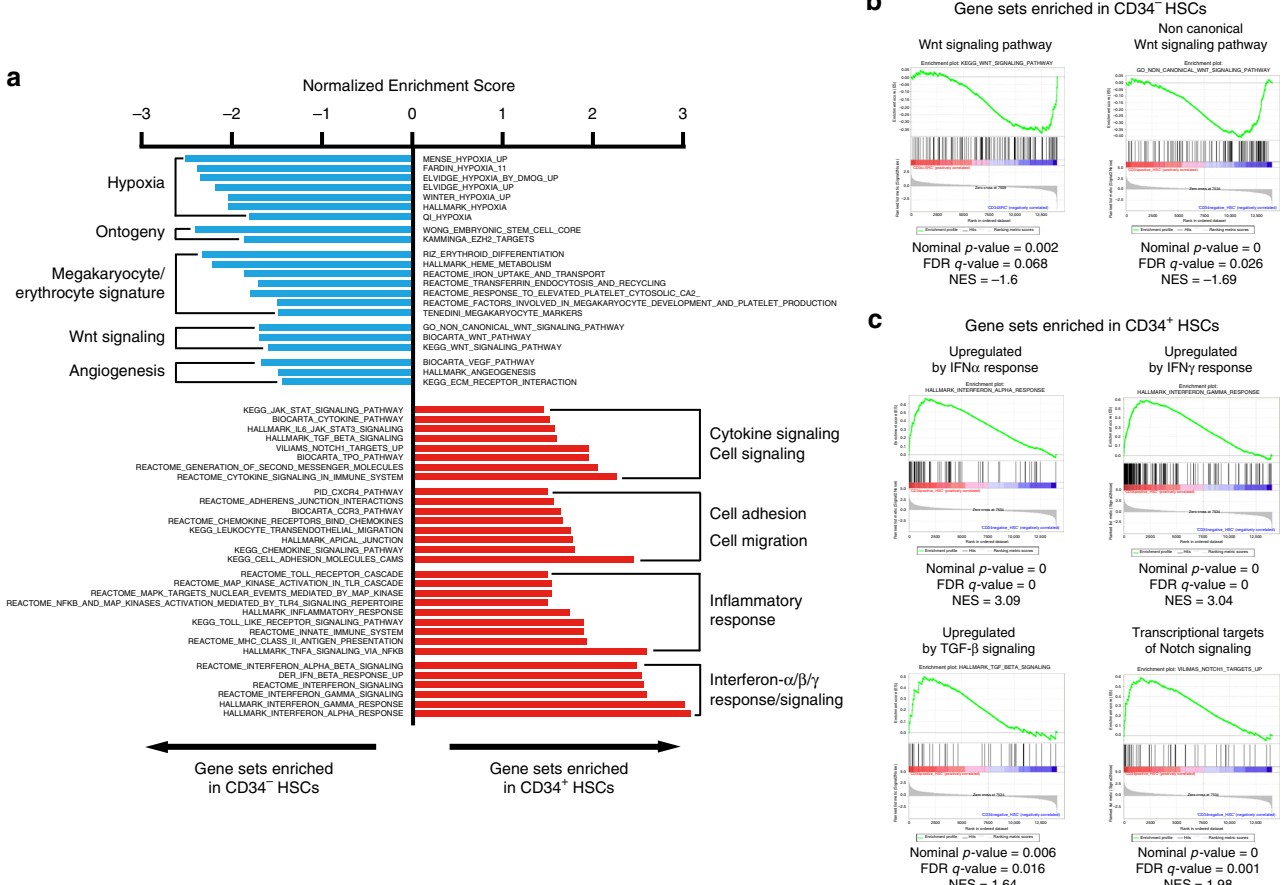

**Fig. 6** A comparison of the gene expression profiles between CD34$^+$ and CD34$^-$ SRCs (HSCs) by a microarray. **a** The gene expression profiles of CD34$^+$ and CD34$^-$SRCs (HSCs) were compared by a Gene Set Enrichment Analysis (GSEA) using the data from the microarray analysis. The gene sets enriched in CD34$^-$ SRCs (blue bar) and those in CD34$^+$ SRCs (red bar) with a Normalized Enrichment Score <−1 or >1 were selected and presented. Detailed lists of the genes in each gene set depicted in the figure can be found in the GSEA Molecular Signatures Database (http://software.broadinstitute.org/gsea/msigdb/index. jsp). **b**, **c** The gene expression profiles that related to the signaling pathways for the HSC maintenance were precisely analyzed. **b** Enrichment plot of the gene sets (presented in **a**) which were enriched in CD34$^-$ HSCs and **c** in CD34$^+$ HSCs are depicted

engraftment potentials without any accessory cells in recipient NOG/NSG mice, indicating that unidentified mouse niche cells/factors can support human HSCs, at least in the initial phase of xenotransplantation, across the species barrier. This is an important issue that should be addressed in the future in order to improve the efficiency of single-cell transplantation and to further clarify the HSC-supporting mechanism.

Single-cell-based gene expression analyses, such as the PCA analysis and unsupervised hierarchical clustering analysis, demonstrated that gene expression profiles of individual CD34$^+$ and CD34$^-$ HSCs were clearly different (Fig. 4a, b). However, immunophenotypically homogeneous CD34$^-$ (18Lin$^-$CD34$^-$CD133$^+$GPI-80$^+$) and CD34$^+$ (18Lin$^-$CD34$^+$CD38$^-$CD133$^+$GPI-80$^+$) HSCs are still very heterogeneous from the point of gene expression. These findings may reflect the variable hematopoietic repopulation patterns of individual CD34$^-$ and CD34$^+$ SRCs (HSCs) observed in the BM, PB and spleen (Supplementary Data 3a to c). As shown in Fig. 5b, individual CD34$^-$ HSCs showed unique gene expression profiles, including for the *IFITM1*, *MPL*, *IKZF1*, *ETV6*, *ALDH1A1*, *IGF1R*, *EZH2* and *MYB* genes, in comparison to CD34$^+$ HSCs. Furthermore, GSEAs also demonstrated clear differences between these two classes of CD34$^+$ and CD34$^-$HSCs (Fig. 6). Among them, we focused on the expression of *IFITM1*. The expression of *IFITM1* in individual CD34$^-$ HSCs was extremely low in comparison to CD34$^+$

HSCs (Fig. 5b). In addition, the GSEA demonstrated that gene sets related to IFN-α/β/γ response/signaling are enriched in CD34$^+$ HSCs. As expected, genes related to IFN signaling, including *STAT1*, *IFITM1*, *IFITM3*, *DDX58*, *IFI44*, *CXCL10* and *CXCL11*, showed significantly higher expression in CD34$^+$ HSCs than in CD34$^-$ HSCs (Supplementary Fig. 11). It is well-documented that type I IFNs are essential for establishing the host antiviral state. However, their roles in human steady-state hematopoiesis have not been elucidated. Our present data suggest that CD34$^-$ HSCs may be more resistant than CD34$^+$ HSCs to chronic activation of the IFN-α pathway. This suggests that CD34$^-$ HSCs may be able to keep their dormancy and survive under the emerging condition. Taken together, these results suggest that independent molecular backgrounds control self-renewal, maintenance and epigenetic regulation in these two classes of CD34$^+$ and CD34$^-$HSCs. The application of the current ultra-high-resolution purification method may provide a clue for uncovering fundamental molecular mechanisms that control human CB-derived CD34$^+$ and CD34$^-$HSCs.

Over the past two decades, the road map of murine HSC differentiation has been reevaluated due to the accumulation of new findings in this field[51–53]. However, that of humans has remained relatively unchanged. Under the current model[1] (Fig. 7a), the immunophenotype of LT-HSC is Lin$^-$CD34$^+$CD38$^-$CD45RA$^-$CD90$^+$CD49f$^+$[3]. These LT-HSCs give rise to MPPs, MLPs,

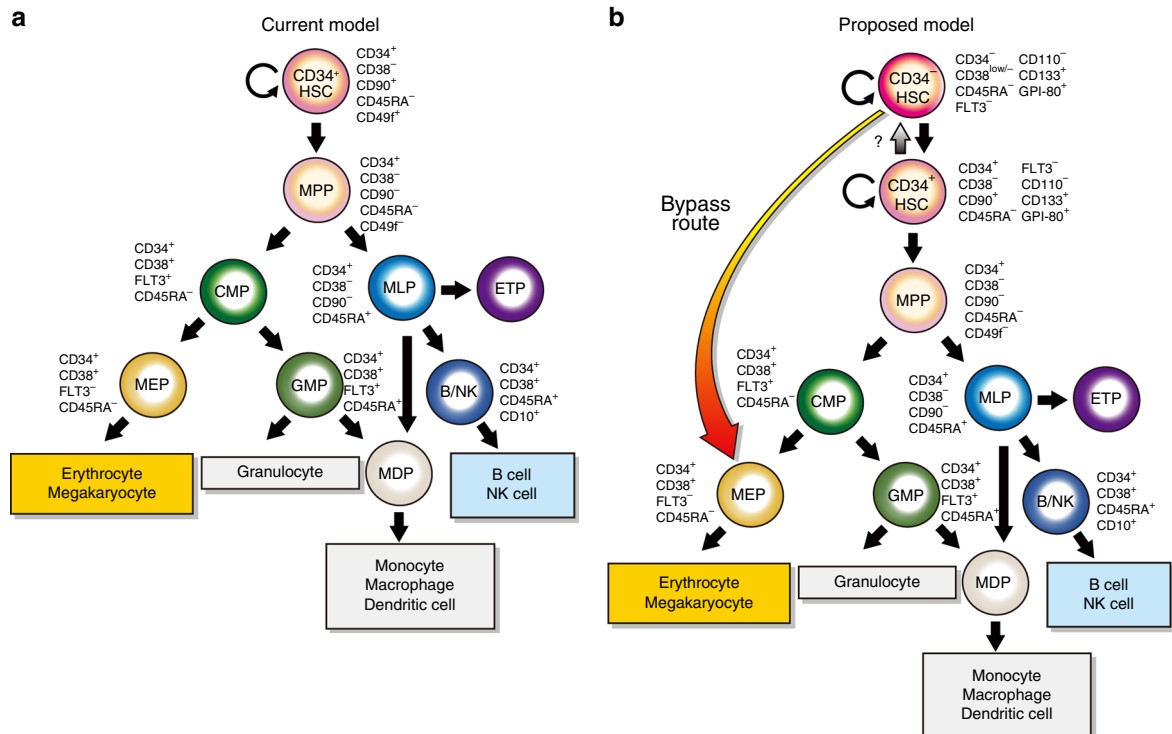

**Fig. 7** The current and proposed models for human HSC hierarchy. **a** The current model[1–3] for the human HSC hierarchy. CD34+ HSCs as defined by a CD34+CD38−CD45RA−CD90+CD49f+ immunophenotype differentiate into MPPs, CMPs, MLPs, GMPs and MEPs. **b** The model[19,20] proposed based on our series of studies[18,21,22,25,33–36,60], in which CD34− HSCs are defined by a CD34−CD38low/−CD45RA−FLT3−CD110−CD133+GPI-80+ immunophenotype. CD34− HSCs generate CD34+ HSCs in vitro[25,35,36] and in vivo[18,33], suggesting that CD34− HSCs reside at the apex of human HSC hierarchy. CD34− HSCs, then, differentiate into MPPs, CMPs, GMPs and MEPs according to the current model[1–3]. Incorporating the present studies, a revised road map, which allows a commitment/differentiation pathway of CD34− HSCs directly into MEPs (bypass route), is shown. MPP: multipotent progenitor, MLP: multilymphoid progenitor, CMP: common myeloid progenitor, GMP: granulocyte/macrophage progenitor, MEP: megakaryocyte/ erythrocyte progenitor

CMPs, GMPs and MEPs[1–3] which are basically close to the mouse model. The findings concerning HSC biology obtained in mice cannot, however, always be extrapolated to humans. Therefore, it is necessary to perform validation studies using human HSCs. Doulatov et al. reported a revised road map for the human HPC hierarchy[2]. They showed that monocytes/macrophages and dendritic cells arise in early lymphoid specification. As very recently reported by Notta et al., human fetal liver contained oligopotent progenitors, while few oligopotent progenitors were present in adult BM that was proposed by the precise analysis of single purified CD34+ HSCs[54].

In the present study, we analyzed the lineage potentials of highly purified human CB-derived CD34+ and CD34−HSCs in vitro as well as in vivo. Single-purified CD34+ HSCs mainly formed CFU-Mix colonies in addition to CFU-GM and BFU-E. In contrast, single-purified CD34− HSCs mainly formed CFU-EM colonies. Unexpectedly, 90% of the colonies formed by 34−133+80− cells were CFU-EM (Fig. 1j). When we transplanted CD34− HSCs into NOG/NSG mice, they were able to produce CD34+ HSCs in vivo, as previously reported[18,33] and gave rise to all types of hematopoietic cells (Figs. 2 and 3). When we co-cultured CD34− HSCs with DP MSCs[35] in the presence of a cocktail of cytokines, they produced CD34+ HSCs in vitro (Supplementary Fig. 10 and Supplementary Data 4) as previously reported[25,35,36]. After co-culturing, the recovered cells were re-cultured in methylcellulose to confirm their lineage potentials, and all types of CFCs, including CFU-GM, BFU-E and CFU-Mix, were detected[35]. These results suggest that CD34−HSCs can generate CD34+ HSCs in the presence of DP MSCs (niche cells)

and give rise to all types of hematopoietic progenitor cells, as observed in xenotransplantation (Supplementary Fig. 7). As shown in Supplementary Fig. 14, the violin plot and heatmap revealed that the gene expression profiles of CD34+ and CD34− HSCs and MEPs were clearly different. The MEPs expressed higher levels of the *NFE2*, *GFI1B*, *GATA2*, *TFRC*, *MYC* and *MYB* genes than CD34+ and CD34− HSCs. Interestingly, CD34− HSCs expressed higher levels of the *MYB* gene, which regulates the biphenotypic fate of human MEPs[55], than CD34+ HSCs. It is true that the gene expression profiling solely is not sufficient to argue for the molecular implication in the difference in self-renewal, maintenance and epigenetic regulation of our currently identified HSC subpopulations. We, however, believe that the difference in the gene expression profiling between the HSC subpopulations provides an important clue for elucidating the molecular background sustaining the differential cellular functions of these HSC subpopulations.

Collectively, in vivo, CD34− HSCs can differentiate in line with our proposed model (Fig. 7b). They first generate CD34+ HSCs, which then give rise to MPPs, MLPs, CMPs, GMPs and MEPs. In contrast, in methylcellulose cultures, CD34− HSCs may bypass the stage of CD34+ HSCs and directly differentiate into MEPs. Consistent with this hypothesis, CD34− HSCs expressed a higher level of gene sets related to megakaryocyte/erythrocyte differentiation (Fig. 6a) and displayed stronger megakaryocyte/erythrocyte differentiation capability even in vivo (Supplementary Fig. 8a). Based on the present studies, we propose a revised megakaryocyte/erythrocyte bypass route as shown in Fig. 7b. However, it remains as a possibility that CD34− HSCs are able to

differentiate via CD34$^+$ HSCs in line with our proposed model under influence of the specific niche cells/factors as seen in vivo.

As shown in Supplementary Fig. 15, we did not detect 18Lin$^-$CD34$^-$CD133$^+$GPI-80$^+$ cells in adult BM cells. These results suggest that CD34$^-$ HSCs may be a CB-specific population, while there is a possibility that young adults' or childrens' BMs contain CD34$^-$ HSCs.

In conclusion, our present study has major implications for the precise characterization of the most primitive class of human CD34$^-$ HSCs as well as for HSCT in transplantation medicine. From the viewpoint of HSCT, it is important to develop an efficient method for expanding CB-derived HSCs for transplantation, as CB transplantation (CBT) is now one of the most commonly used sources for allogeneic HSCT worldwide[42–44]. However, the number of HSCs in CB units is limited, so CBT is often associated with delayed neutrophil and platelet recovery as well as graft failure, which represents a life-threatening complication[42–44]. In this study, we tried to expand very primitive CB-derived CD34$^-$ SRCs (HSCs) (18Lin$^-$CD34$^-$CD133$^+$GPI-80$^+$ cells) using SCF, TPO, FL plus StemRegenin-1 (SR-1)[45] or pyrimidoindole derivative (UM171)[46]. However, it was hard to expand CD34$^-$ and CD34$^+$ SRCs (HSCs) under these culture conditions (Supplementary Fig. 13). Further studies will be required to develop an efficient method for HSC ex vivo expansion.

## Methods

**Collection of CB and bone marrow samples**. CB samples were obtained from normal full-term deliveries with written informed consent. Human BM samples were obtained from healthy donor BM remaining in the collection bag after clinical BM transplantation with informed consent. This study was approved by the Institutional Review Board of Kansai Medical University. The CB- and BM-derived lineage-negative (Lin$^-$) mononuclear cells were separated using an EasySep Human Progenitor Cell Enrichment Kit (StemCell Technologies, Vancouver, BC, Canada) in accordance with the manufacturer's instructions, as previously reported[18,21,22,25,33–36].

**Immunostaining of Lin$^-$ cells for flow cytometry and sorting**. The pooled abovementioned CB-derived Lin$^-$ cells from multiple donors were stained with various monoclonal antibodies (mAbs) (Supplementary Data 5) for 30 min at 4 °C in Ca$^{2+}$- and Mg$^{2+}$-free phosphate-buffered saline (PBS$^-$) (Nacalai Tesque, Kyoto, Japan) containing 2% fetal calf serum (FCS) (Biofill, Elsternwick Victoria, Australia) (PBS$^-$/FCS). We used fluorescein isothiocyanate (FITC)-conjugated 18 Lin mAbs, against CD2, CD16, CD24 and CD235a (DAKO, Kyoto, Japan); CD3, CD7, CD10, CD11b, CD20, CD41 and CD66c (Beckman Coulter, Fullerton, CA, USA); CD19 and CD56 (BD Biosciences, San Jose, CA, USA); CD4, CD14, CD33 and CD127 (eBioscience, San Diego, CA, USA); CD45RA (Southern Biotech, Birmingham, AL, USA); a Brilliant Violet 510 (BV510)-conjugated anti-CD45 mAb (BioLegend, San Diego, CA, USA); a Brilliant Violet 421 (BV421)-conjugated anti-CD34 mAb (BioLegend), apophycocyanin (APC)-conjugated anti-CD133/1 mAb (Miltenyi Biotec, Bergish Gladbach, Germany), a phycoerythrin (PE)-conjugated anti-GPI-80 mAb (MBL Medical & Biological Laboratories, Nagoya, Japan) and a PE-Cy7-conjugated anti-CD38 mAb (BioLegend). The cells were then washed once with PBS$^-$/FCS, and resuspended in a 7-amino-actinomycin D (7-AAD) (Beckman Coulter)-containing PBS$^-$/FCS solution before the flow cytometric (FCM) analyses or FACS. The stained cells were then sorted into four fractions, including 18Lin$^-$CD34$^+$CD38$^-$CD133$^+$GPI-80$^{+/-}$ and 18Lin$^-$CD34$^-$CD133$^+$GPI-80$^{+/-}$ cells, using a FACSAriaII and a FACSAriaIII (BD Biosciences) (Fig. 1a–f). The purity of sorted cells was always confirmed by a post-sorting purity analysis. In order to define the CD34$^-$ fraction for flow cytometric analyses, the gate was set based on the Fluorescent Minus One (FMO) control.

**The surface marker expression of purified HSC populations**. The expression of CD90, CD49f, CD93 and CD184 (CXCR4) on the surface of 18Lin$^-$CD34$^+$CD38$^-$CD133$^+$GPI-80$^+$ and 18Lin$^-$CD34$^-$CD133$^+$GPI-80$^+$ cells was analyzed using FACSCantoII (BD Biosciences). Human CB-derived Lin$^-$ cells obtained from individual samples ($n = 7$) were stained with a mixture of FITC-conjugated 18 Lin mAbs, PE-conjugated anti-GPI-80 mAbs, PE-Cy7-conjugated anti-CD38 mAb, APC-conjugated anti-CD133/1 mAb, APC-Cy7-conjugated anti-CD34 mAb (BioLegend), BV510-conjugated anti-CD45 mAb and biotin-conjugated anti-CD90 (BioLegend), CD49f (BioLegend), CD184 (BioLegend) or CD93 mAb (OriGene). Biotin-conjugated mAbs were then reacted with BV421-conjugated streptavidin for 30 min at 4 °C. The samples were washed twice with PBS$^-$/FCS followed by 7-AAD

staining before the analysis. The positive gate for the BV421 fluorescence channel was defined according to the FMO control.

**Isolation of target cells for single-cell analyses**. In order to analyze the single-cell gene expression profiles, CB-derived single CD34$^+$ HSCs and CD34$^-$ HSCs were isolated by FACSAriaII with the automated cell deposition unit (ACDU). Single CD90$^+$ HSCs, MPPs, multi-lymphoid progenitors (MLPs), CMPs, granulocyte and monocyte progenitors (GMPs), megakaryocyte and erythrocyte progenitors (MEPs), erythroblasts, monocytes/macrophages, megakaryocytes, B cells and T cells were also isolated in accordance with previous reports[2,3] by FACS. The details of the immunophenotypes of these fractions of cells are described in Supplementary Data 6.

**An analysis of the morphology of the highly purified cells**. Portions of 18Lin$^-$CD34$^+$CD38$^-$CD133$^+$GPI-80$^{+/-}$ and 18Lin$^-$CD34$^-$CD133$^+$GPI-80$^{+/-}$ cells were cytospun onto a slide glass. They were then stained with May–Grunwald–Giemsa for morphological analysis. The images of the cells were recorded using an inverted microscope system (BZ-9000, Keyence, Osaka, Japan) with a ×100 Apo oil objective lens (Fig. 1g).

**An analysis of colony-forming capacity of single cells**. Single 18Lin$^-$CD34$^+$CD38$^-$CD133$^+$GPI-80$^{+/-}$ or 18Lin$^-$CD34$^-$CD133$^+$GPI-80$^{+/-}$ cells were sorted into a 96-well flat-bottomed plate (Corning Inc., Corning, NY, USA) by the FACSAriaII with the ACDU, as reported[56]. Each well contained 50 μL of our standard methylcellulose culture medium containing 30% FCS (Biosera, Kansas, MO, USA) and 6 cytokines (thrombopoietin [TPO], stem cell factor [SCF] [R&D Systems, Minneapolis, MN, USA], interleukin-3 [IL-3], granulocyte/macrophage colony-stimulating factor [GM-CSF], granulocyte colony-stimulating factor [G-CSF] and erythropoietin [Epo]), as reported[21,22,34,53]. TPO, GM-CSF, G-CSF, IL-3 and Epo were kindly provided by Kyowa Hakko Kirin Company (Tokyo, Japan). The plates were cultured for 10 to 14 days and then assayed as reported previously[21,22,33,35,36,56]. We analyzed various types of colony-forming cells (CFCs), including megakaryocyte-containing CFCs, as reported[21,22,33,35,36,56]. We picked up all of the colonies derived from single-cells under an inverted microscope. Cytospin preparations were made using Cytospin (Thermo Fisher Scientific, Waltham, MA, USA) for all colonies picked up from a total of eight 96-well flat-bottomed plates. These cytospin preparations were stained with May–Grunwald–Giemsa for a morphological analysis. We identified neutrophils (n), macrophages (m), eosinophils (e), erythroblasts/erythrocytes (E) and megakaryocytes (M) and classified each colony as a single and/or multi-lineage one. Each colony type was classified as follows: colony-forming unit-granulocyte (CFU-G) containing n; colony-forming unit-macrophage (CFU-M) containing m; colony-forming unit-granulocyte/macrophage (CFU-GM) containing nm, ne or nme; colony-forming unit-eosinophil (CFU-Eo) containing e; burst-forming unit-erythroid (BFU-E) containing E; colony-forming unit-megakaryocyte (CFU-Meg) containing M; colony-forming unit-erythrocyte/megakaryocyte (CFU-EM) containing E and M; colony-forming unit-mixed (CFU-Mix) containing nE, nM, mM, nmE, nEM, nmM, neE, neM, nmEM or nmeEM (Supplementary Fig. 2).

*Single-cell proliferation assay*. Single 18Lin$^-$CD34$^+$CD38$^-$CD133$^+$GPI-80$^+$, 18Lin$^-$CD34$^-$CD133$^+$GPI-80$^+$ and 18Lin$^-$CD34$^+$CD38$^{high}$ cells were sorted using the ACDU into 60-well MiniTray (Thermo Fisher Scientific) which contained StemPro-34 SFM plus GlutaMax (1:100, Thermo Fisher Scientific) supplemented with a cocktail of 2 cytokines (100 ng/mL TPO and 50 ng/mL SCF: referred to as basal medium), or with 10% FCS (Biosera) added and supplemented with the a cocktail of 7 cytokines (100 ng/mL TPO, 50 ng/mL SCF, 50 ng/mL fms-like tyrosine kinase-3 ligand (FL) [Cell Signaling Technology, Beverly, MA, USA], 20 ng/mL of G-CSF, 10 ng/mL of GM-CSF, 10 ng/mL IL-3 and 10 ng/mL IL-6 [Peprotech, Rocky Hill, NJ, USA]; referred to as complete medium) at 37 °C under 5% CO$_2$ in air. The first and second cell division times were assessed by microscopic observation at 12 h intervals from 30 to 144 h.

*Cell cycle analyses*. The cell cycle status of the abovementioned CB-derived highly purified HSC populations, such as 18Lin$^-$CD34$^+$CD38$^-$CD133$^+$GPI-80$^{+/-}$ and 18Lin$^-$CD34$^-$CD133$^+$GPI-80$^{+/-}$ cells were analyzed as mentioned below. As controls, the cell cycle status of CB- and BM-derived CD90$^+$ HSCs, MPPs, MLPs and Lin$^-$CD34$^+$CD38$^{high}$ cells was also analyzed. These hematopoietic stem/progenitor cell (HSPCs) were obtained by FACS as follows; cryopreserved CB and BM samples were thawed and Lin$^-$ cells were immediately separated using an EasySep Human Progenitor Cell Enrichment Kit as described above. Immunomagnetically enriched Lin$^-$ cells were stained with surface markers by a cocktail of fluorochrome-conjugated mAbs as shown in Supplementary Data 5. Intracellular staining was then carried out using a BD Cytofix/Cytoperm Kit (BD Bioscience) in accordance with the manufacturer's instruction. The cells were fixed with Cytofix/Cytoperm buffer (BD Bioscience) for 10 min at 4 °C. Subsequently, the cells were placed into PermWash buffer (BD Bioscience) and then reacted with PE-Cy7-conjugated anti-human Ki-67 mAb (1:25, BioLegend) and 7-AAD (1:50) at 4 °C for overnight. The expressions of surface markers and Ki-67 antigens and the DNA

content in each fraction of cells were analyzed by an eight-color FCM (FACSCanto II). $G_0$, $G_1$ and $S/G_2/M$ gates were defined according to the FMO control.

**NOG and NSG mice.** Female six-week-old NOD.Cg-Prkdc[scid]Il2rg[tm1Sug]/Jic (NOG)[57] and NOD.Cg-Prkdc[scid]Il2rg[tm1Wjl]/SzJ (NSG)[58] mice were purchased from the Central Institute of Experimental Animals (Kawasaki, Japan) and Charles River Laboratories Japan, Inc. (Yokohama, Japan), respectively. All mice were handled under sterile conditions and maintained in germ-free isolators in the Central Laboratory Animal Facilities of Kansai Medical University. In order to reduce the oxidative stress, mice were given feed containing 1.3% N-acetyl-L-cysteine (NAC) for the entire repopulation period, as previously reported[59]. The animal experiments were approved by the Animal Care Committees of Kansai Medical University. NOG and NSG mice (8-week-old) were sub-lethally irradiated (2.5 Gy using a [137]Cs-γ irradiator) immediately before all of the transplantation studies.

**Intra-BM injection of purified cells.** IBMI was carried out as reported previously[18,21,22,25,33–36,60]. Briefly, after sterilization of the skin around the left knee joint, the knee was flexed to 90 degrees and the proximal side of the tibia was drawn to the anterior. A 26-gauge needle was inserted into the joint surface of the tibia through the patellar tendon and then inserted into the BM cavity. Using a Hamilton microsyringe, the specified number of cells per 10 µL of α-medium plus 5% FCS were carefully injected from the bone hole into the BM cavity.

**SCID-repopulating cell assay by FCM.** The SRC assays were performed by the previously reported method[18,21,22,25,34–36,60]. In this study, a specified number of 18Lin⁻CD34⁺CD38⁻CD133⁺GPI-80[+/−] and 18Lin⁻CD34⁻CD133⁺GPI-80[+/−] cells were transplanted by IBMI into eight-week-old NOG or NSG mice. Human hematopoietic repopulations in the mouse BMs were serially analyzed 12 and 18 weeks after transplantation by the BM aspiration method[18,21,22,25,34–36,60]. The aliquots of BM aspirates were stained with a Pacific Blue (PB)-conjugated anti-human CD45 mAb (BioLegend), a PE-Cy7-conjugated anti-mouse CD45.1 mAb (Beckman Coulter), FITC-conjugated anti-human CD19 mAb, PE-conjugated anti-human CD33 mAb (Beckman Coulter), APC-conjugated anti-human CD34 mAb (BD Biosciences) and 7-AAD and were analyzed by six-color FCM (FACS CantoII). At weeks 20 to 22, the mice were killed, and the BM cells were collected by crushing the femurs and tibiae from each of mice in a mortar. The cells from the PB, spleen (SP) and thymus (Thy) of these mice were also collected. The repopulation of human hematopoietic cells in the murine BM, PB, SP and Thy were determined by detecting the number of 7-AAD⁻ cells positively stained with PB-conjugated anti-human CD45 mAb by six-color FCM (FACS CantoII). The cells were also stained with a PE-Cy7-conjugated anti-mouse CD45.1 mAb (Beckman Coulter); FITC- or BV510-conjugated anti-human CD3 (BM, Thy), CD19 (BM, PB, SP), CD11b (BM) and CD235a (BM) mAbs; PE-conjugated anti-human CD4 (Thy) (eBioscience), CD33 (BM, PB, SP), CD14 (BM) and CD41 (BM, SP) mAbs (Beckman Coulter), and APC-conjugated anti-human CD8 (Thy) (eBioscience) and CD34 (BM, PB, SP) mAbs (BD Biosciences) for the detection of human stem/progenitor, B-lymphoid, T-lymphoid and myeloid/monocytic hematopoietic cells. For the precise analysis of the NK-cell development in the spleen, the cells were stained with an FITC-conjugated anti-human CD56 mAb (BD Biosciences). In order to determine the human CD45⁺ gate for the transplantation analyses, BM tissue from NSG mice that did not receive human cells was used as a negative control and human cord blood-derived mononuclear cells were used as a positive control. In all transplantation experiments, the mice were scored as positive if more than 0.01% of the total murine BM cells were human CD45⁺ cells, as previously reported[22].

**Secondary transplantation.** For secondary transplantations, murine BM cells were obtained 19 to 23 weeks after the transplantation from pairs of the femurs and tibiae from engrafted primary recipient NOG/NSG mice receiving the above-mentioned purified 18Lin⁻CD34⁺CD38⁻CD133⁺GPI-80 ⁺ and 18Lin⁻CD34⁻CD133⁺GPI-80⁺ cells. The secondary transplantation was then performed, as previously reported[18,21,22,33,36]. Briefly, 1/5 of the whole BM cells were transplanted into both tibiae of secondary recipient mice by IBMI. Some of the whole BM cells were stained with the FITC-conjugated 18 Lin mAbs, BV510-conjugated anti-CD45 mAb, BV421-conjugated anti-CD34 mAb, PE-Cy7-conjugated anti-CD38 mAb, APC-conjugated anti-CD133/1 mAb and PE-conjugated anti-GPI-80 mAb. The 18Lin⁻CD34⁺ and 18Lin⁻CD34⁻cells were then resorted and transplanted into secondary recipient mice. Expression of CD38, CD133 and GPI-80 on these 18Lin⁻CD34[+/−] cells recovered from the engrafted primary recipient mice was analyzed by FCM. Finally, at 19 to 23 weeks after transplantation, presence of human CD45⁺ cells in the secondary recipients' BMs was analyzed by FCM, as described above.

**In vivo limiting dilution analysis.** To assess frequency of SRCs in CB-derived 18Lin⁻CD34⁺CD38⁻CD133⁺GPI-80[+/−] and 18Lin⁻CD34⁻CD133⁺GPI-80[+/−] cells, various numbers of cells, the precise numbers of which are shown in Supplementary Data 1, were transplanted into NSG mice by IBMI, as previously

reported[18,21,22,25,33]. The human cell repopulation in the mouse BM was analyzed at 20 weeks after transplantation by FCM.

**Single-cell transplantation.** Single 18Lin⁻CD34⁺CD38⁻CD133⁺GPI-80⁺ or 18Lin⁻CD34⁻CD133⁺GPI-80⁺cells were sorted into a 60-well MiniTray in 20 µL of α-medium with 10% FCS using the ACDU with a FACSAriaII. After sorting, the cells were incubated in the CO₂ incubator for 1 h, and visualized under an inverted microscope. Each cell was transferred into a Hamilton microsyringe under the inverted microscope. Wells were re-visualized to confirm that the cell had been successfully transferred into the syringe. In another independent experiment, we confirmed the viability (>99%) of single-sorted cells by Trypan blue dye exclusion. We next analyzed efficiency of our single-cell transplantation method. We sorted single 18Lin⁻CD34⁺CD38⁻CD133⁺GPI-80⁺ or 18Lin⁻CD34⁻CD133⁺GPI-80⁺ cells into a 60-well MiniTray. After transferring the cells into a Hamilton microsyringe, the cells were expunged into a new 60-well MiniTray and spun down and visualized. We observed a single cell in 87% of the wells ($n = 117$), indicating that only 87% of recipient mice transplanted with single 18Lin⁻CD34⁺CD38⁻CD133⁺GPI-80⁺ or 18Lin⁻CD34⁻CD133⁺GPI-80⁺ cells actually received a single cell. This efficiency was taken into consideration in our evaluation of the single-cell transplantation data of LDA.

**Single-cell initiated serial transplantation.** We performed secondary transplantations from primary engrafted recipient mice that received single purified 18Lin⁻CD34⁺CD38⁻CD133⁺GPI-80⁺ and 18Lin⁻CD34⁻CD133⁺GPI-80⁺ cells at 21 to 23 weeks after transplantation. In these cases, we transplanted 2/5 of the whole BM cells into both tibiae of two secondary recipient mice by IBMI. Tertiary transplantation was performed in the same way as secondary transplantations; we transplanted 2/5 of the whole BM cells obtained from secondary recipient mice into both tibiae of two tertiary recipient mice by IBMI. The human CD45⁺ cell repopulation in the secondary and tertiary recipient mice was analyzed at 12 weeks after the transplantation by the BM aspiration technique. At 21 to 23 weeks after transplantation, the mice were killed and the BM cells were collected by crushing the femurs and tibiae from each of the mice in a mortar. The cells from the PB, SP and Thy of these mice were also collected. The repopulation of human hematopoietic cells in the murine BM, PB, SP and Thy were determined by detecting the number of 7-AAD⁻ cells positively stained with PB-conjugated anti-human CD45 mAb by six-color FCM (FACS CantoII). The cells were also stained with a PE-Cy7-conjugated anti-mouse CD45.1 mAb (Beckman Coulter); FITC- or BV510-conjugated anti-human CD3 (BM, Thy), CD19 (BM, PB, SP), CD11b (BM) and CD235a (BM) mAbs; PE-conjugated anti-human CD4 (Thy) (eBioscience), CD33 (BM, PB, SP), CD14 (BM) and CD41 (BM) mAbs (Beckman Coulter), and APC-conjugated anti-human CD8 (Thy) (eBioscience) and CD34 (BM, PB, SP) mAbs (BD Biosciences) for the detection of human stem/progenitor, B-lymphoid, T-lymphoid and myeloid/monocytic hematopoietic cells. For the precise analysis of the NK-cell development in the spleen, the cells were stained with an FITC-conjugated anti-human CD56 mAb (BD Biosciences).

**Real-time reverse transcription polymerase chain reaction.** The primers were designed using the NCBI/Primer BLAST (http://www.ncbi.nlm.nih.gov/tools/primer-blast/) and Primer3 web tool (http://primer3.ut.ee/). The primer pairs used for RT-PCR are shown in Supplementary Data 7. The primer melting temperature (Tm) of all primers was set at 59 to 63 °C. We designed both the first (for pre-amplification) and second (for real-time PCR) primer pairs for individual target genes. Both the forward and reverse second primers were designed to slide at least three bases inside of the amplicon produced by the first primer pairs.

To validate the gene amplification by each primer pair, total RNA was extracted from CB-derived mononuclear cells using an RNeasy Plus Micro kit (QIAGEN, Hilden, Germany). Reverse transcription was then carried out using an iScript cDNA synthesis kit (Bio-Rad Laboratories, Hercules, CA, USA) as follows: 5 min at 25 °C, 30 min at 42 °C and 5 min at 85 °C. The cDNA was then mixed with Power SYBR green Master Mix (Thermo Fisher Scientific), and real-time PCR were then carried out by Rotor-Gene Q (QIAGEN). PCR amplicons were subjected to gel electrophoresis, and the size of the PCR amplicon was validated by comparison with the predicted product size, which was calculated using NCBI/Primer BLAST and Primer3 web tools.

To compare the expression of the VNN2 gene between 18Lin⁻CD34⁺CD38⁻CD133⁺GPI-80[+/−] and 18Lin⁻CD34⁻CD133⁺GPI-80[+/−] cells, total RNA was extracted from $1.4 \times 10^4$ of each fraction of cells. As positive and negative controls, $1.4 \times 10^4$ cells of CB-derived neutrophils and THP-1 cells were used. Reverse transcriptions and real-time PCR were then carried out as described above. The primer sequences for the VNN2 genes were as follows: forward primer (5'-TGGACGGGCCAGTAGAAACTGC-3'), reverse primer (5'-ACCAAACGCCCATCTTTCAGC-3'). The expression of VNN2 was normalized by that of ACTB (β-actin) in each sample. The relative gene expression was calculated by using the $2^{-\Delta\Delta C_T}$ method. All primers were purchased from Thermo Fisher Scientific.

**Cell line.** THP-1 cell line was purchased from RIKEN BRC CELL BANK (Tsukuba, Japan).

**Single-cell gene expression analyses**. First, single 18Lin⁻CD34⁺CD38⁻CD133⁺GPI-80⁺ or 18Lin⁻CD34⁻CD133⁺GPI-80⁺ cells were sorted by FACS into 96-well PCR plates (Thermo Fisher Scientific) containing 9 μL of Cells Direct One-Step qPCR master mixture (Thermo Fisher Scientific) combined with a mixture of pre-amplification primers (listed in Supplementary Data 7). The cells were then immediately lysed by using a vortex mixer. Reverse transcription was carried out for 15 min at 50 °C and 2 min at 95 °C. Subsequently, pre-amplification was carried out for 20 cycles as follows: 15 s at 95 °C and 4 min at 60 °C. Pre-amplified cDNA were stored at −80 °C until immediately before the next step, when it was treated with 4 U/μL of Exonuclease I (New England BioLab Inc., Ipswich, MA, USA) for 30 min at 37 °C followed by treatment for 15 min at 80 °C. The diluted pre-amplified cDNA was loaded onto 96.96 DynamicArray chips (Fluidigm, South San Francisco, CA, USA) and mixed with Assay Loading Reagent (Fluidigm) and mixture of target gene-specific nested primers (listed in Supplementary Data 7). Single-cell real-time PCR was then performed using a Biomark HD System (Fluidigm) in accordance with the manufacturer′s instruction. Real-time PCR was carried out at 70 °C for 40 min, 60 °C for 30 s and 95 °C for 1 min followed by 35 cycles of amplification (96 °C for 5 s and 60 °C for 20 s). Ct values were calculated using the BioMark HD System Real-time PCR Analysis software program (Fluidigm) and filtered according to a set of quality control rules.

To confirm the presence of cells, we used four reference genes: *PGK1*, *PTPRC*, *ACTB* and *GAPDH*. The cells with Ct > 24 cycles for *PGK1* and *PTPRC* and >20 cycles for *ACTB* and *GAPDH*, were excluded from the data. In addition, the CD34⁺ HSCs and CD90⁺ HSCs with Ct values >24 cycles for *CD34* were also excluded from the data. Other genes with Ct values >24 cycles in all target cells were excluded from the data. This resulted in a single-cell gene expression data set (83 genes including 4 reference genes) consisting of 126 cells (33 18Lin⁻CD34⁺CD38⁻CD133⁺GPI-80⁺ cells, 23 18Lin⁻CD34⁻CD133⁺GPI-80⁺ cells, 5 CD90⁺ HSCs, 9 MPP, 8 CMP, 6 MLP, 7 MEP, 11 GMP, 5 erythroblasts, 5 megakaryocytes, 4 T cells, 4 B cells and 6 monocytes/macrophages) from a total of 2 chip runs. These experiments were performed independently three times. The data were analyzed using the SINGuLAR Analysis Toolset software program (Fluidigm). A PCA and unsupervised hierarchical clustering analysis were performed in accordance with the manufacturer's instruction. PCAs were performed on data from CD34⁺ HSCs, CD34⁻ HSC and CD90⁺ HSCs. Unsupervised hierarchical clustering analyses were performed on data derived from all 126 cells. Four reference genes were excluded from these analyses. The violin plots were depicted using the SINGuLAR Analysis Toolset software program. In additional experiments, we analyzed single-cell gene expression profiles of CD34⁺/⁻ HSCs and various progenitors, including MEPs, as described above.

**Microarray analyses**. Total RNA from $3.4 \times 10^4$ to $5 \times 10^4$ 18Lin⁻CD34⁺CD38⁻CD133⁺GPI-80⁺, 18Lin⁻CD34⁻CD133⁺GPI-80⁺ and 18Lin⁻CD34⁻CD133⁻cells were extracted using an RNeasy Plus Micro kit (QIAGEN). The RNA quality was assessed using the Agilent Model 2100 Bioanalyzer (Agilent Technologies, Palo Alto, CA, USA). Total RNA (1–6.5 ng) was amplified using an Ovation Pico WTA System V2 (NuGEN, San Carlos, CA, USA). The amplified RNA was labeled with Cy3 using a SureTag Complete DNA Labeling Kit (Agilent Technologies). Subsequently, the amplified RNA was hybridized with a SurePrint G3 Human Gene Expression 8 × 60 K v3 Microarray platform (Agilent Technologies), and the data were scanned in accordance with the manufacturer's protocol. In order to normalize the data between each of the samples, the mean signal intensity of the individual samples was adjusted to 2500. A gene set enrichment analysis (GSEA) was also performed in accordance with previous reports[61].

**Ex vivo expansion of CB-derived CD34⁻ HSCs**. 18 Lin⁻CD34⁻CD133⁺GPI-80⁺ cells (2850 cells/well, triplicate culture) were plated onto 24-well culture plate and cultured in StemSpan SFEM (StemCell Technologies) supplemented with a cocktail of cytokines, including 50 ng/mL TPO, 50 ng/mL SCF and 100 ng/mL FL and 10 μg/mL low-density lipoprotein (EDM Millipore, Temecula, CA, USA) as recently reported[45,46]. StemRegenin⁻1 (SR-1) (StemCell Technologies) and Pyrimidoindole derivative (UM171) (StemCell Technologies) were dissolved in dimethyl sulfoxide (DMSO) (Nacalai Tesque). Subsequently, 0.1% DMSO (control), 750 nM SR-1 and 35 nM UM171 were added to the culture medium at the start of these cultures. At Day 12, the cultured cells were collected and stained with a mixture of the following mAbs: FITC-conjugated 12Lin mAbs including CD2, CD3, CD4, CD7, CD8, CD10, CD11b, CD14, CD19, CD20, CD56 and CD235a, a FITC-conjugated anti-CD45RA mAb, a PE-conjugated anti-GPI-80 mAb, PE-Cy7-conjugated anti-CD38 mAb, an APC-conjugated anti-CD133/1 mAb, a BV421-conjugated anti-CD34 mAb and a BV510-conjugated anti-CD45 mAb. 12Lin⁻CD45RA⁻CD34⁺ cells and 12Lin⁻CD45RA⁻CD34⁻ cells produced from 18Lin⁻CD34⁻CD133⁺GPI-80⁺ cells were then sorted and collected by FACSAriaIII. The CD34-negative cell gate was defined according to the FMO control. Twelve Lin⁻ and CD45RA-negative cell gates were also set according to the FMO controls. In order to prevent the loss of CD34⁺ SRCs, the CD34-positive cell gate was set adjacent to the CD34-negative cell gate. Varying numbers of 12Lin⁻CD45RA⁻CD34⁺ cells generated under each culture condition were transplanted into six to eight NSG mice/group by IBMI. All 12Lin⁻CD45RA⁻CD34⁻ cells harvested under each culture condition were split and transplanted into six or seven NSG mice by IBMI. The frequencies of SRC in the culture-generated 12Lin⁻CD45RA⁻CD34⁺ cells were estimated using an ELDA web tool[62], as described in the Statistical analyses. The details of the design of LDA

for cultured cells are described in Supplementary Fig. 13a. At 6 weeks after transplantation, the human multi-lineage hematopoietic repopulation (including CD34⁺, CD19⁺ and CD33⁺ cells) in the BM of the left tibiae of each mouse (injected site) were analyzed by a BM aspiration technique.

**Statistical analyses**. In Fig. 1h and Supplementary Fig. 4c, and Supplementary Fig. 13c, d, and f, differences between each pair of all the means were examined by Tukey's multiple comparison procedure. In Fig. 1i, the differences in the colony-forming efficiency of each cell types were examined by a two-way analysis of variance. In Fig. 2b,c and Supplementary Fig. 11, the difference in the means between each pair was examined using a two-tailed Student's *t*-test. In Supplementary Fig. 4d and Supplementary Fig. 8a and b, the significance of differences was determined using the Mann–Whitney *U* test. Differences were considered statistically significant at a confidence level of <0.05. In Fig. 4a, the regions of the CD34⁺ and CD34⁻ HSCs on the PCA plot were determined by a Fisher's linear discriminant analysis. In the LDA (Supplementary Fig. 5 and Supplementary Fig. 13e), the frequencies of SRCs were calculated as previously reported[62].

**Data availability statement**. We presented microarray data in Fig. 6, Supplementary Fig. 11 and Supplementary Fig. 12. Microarray expression data are available at NCBI GEO: GSE100354.

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

## Acknowledgements

This work was supported by Grants-in-Aid for Scientific Research C (Grant Nos. 21591251, 24591432 and 16K09881) from the Ministry of Education, Culture, Sports, Science and Technology (MEXT) of Japan; a grant from the Science Frontier Program of the MEXT; a grant from the Strategic Research Base Development program for Private Universities from the MEXT; MEXT-Supported Program for the Strategic Research Foundation at Private Universities (S1101034 and S1201038); a grant from the Promotion and Mutual Aid Corporation for Private Schools of Japan; a grant from the Japan Leukemia Research Foundation; a grant from the Mitsubishi Pharma Research Foundation; a grant from the Takeda Science Foundation; a grant from the Terumo Life Science Foundation and a grant from SENSHIN Medical Research Foundation. The authors are grateful to the Japanese Red Cross Kinki Cord Blood Bank for providing CB samples used in this study. Kyowa Hakko Kirin Company (Tokyo, Japan) is also acknowledged for providing the growth factors used in this study.

## Author contributions

K.S. designed and performed a majority of the experiments and analyzed and interpreted the data; Y.M. designed and performed the experiments, analyzed and interpreted the data and contributed to the manuscript writing; H.K., R.N., T.F., provided study material and contributed to some parts of the experiments; H.A., provided advice on the statistical analyses; Y.T., contributed to the data analysis and interpretation; Y.S., conceived of and designed the study, analyzed and interpreted the data, provided financial and administrative support, and wrote the paper.
