## [Peer Review File · Nature Communications]

Reviewers' comments:

Reviewer #1 (Remarks to the Author):

General Comments

In this present manuscript, Sumide, et. al., suggest through an extensive effort that a population of CD34⁻ hematopoietic stem cells (HSCs) is at the apex of hematopoiesis, and that these cells they have now defined by phenotype show megakaryocyte-erythroid differentiation bias potential in an in vitro culture system. While defining the earliest HSC in the human system by phenotype is of interest, there are a number of problems with the studies that the authors will need to adequately address before their interpretations can be considered definitive.

Currently, the HSC and progenitor cell system are most rigorously defined in the mouse bone marrow by phenotype using multiple cell surface antigens. In the mouse system it is clear that the earliest most immature HSC with long-term engrafting capability in lethally irradiated primary mouse recipients with extensive self-renewal capacity as defined by secondary mouse transplants is CD34⁻. While less rigorously defined in the human system, e.g. with cord blood cells, there has been much head-way made in phenotypically defining the human long-term repopulating (e.g. in sublethally-irradiated immune deficient mice using SCID repopulating cells, SRCs) self-renewing HSC. Most investigators consider the human HSC as being CD34⁺, although some investigators, including the present authors have provided evidence that the earliest HSC may be CD34⁻. The present study attempts to further define their belief that the human HSC is CD34⁻, thus bringing this human phenotype in closer line with that of the mouse HSC.

A major problem with the present extensive investigation is the lack of clarity on how rigorously they have defined this CD34⁻ human HSC. Sometimes they refer to studies with CD34⁻ cells, but in many places they refer to a CD34^{+/-} cell population, or a CD34^{low/-} population. This is confusing. On top of this, the manuscript is a hard read and evaluate paper that has so many different phenotypes noted that it becomes a difficult effort to follow all the experiments noted, of which there are many. Also, statistical evaluation is missing from many of the analyses making it unclear if differences are real or not. In addition, the concept that the CD34⁻ cells they are working with have a megakaryocyte-erythroid bias is based entirely on the differentiation of these cells in vitro, as it is not apparent in vivo. Hence, there interpretations that there is a bias in vivo in this differentiation directly at the level of this CD34⁺ cell may not be correct.

In their earlier studies, they found that SRCs were enriched in a Lin⁻CD34⁻CD133⁺ population, and later they reported that CD34⁻ SRCs were also enriched in the Lin⁻CD34⁻GPI-80⁺ cell fraction. In the present study, they combined use of CD133 and GPI-80, and found that CD34⁻ SRCs were further enriched in Lin⁻CD34⁻CD133⁺ GPI-80⁺ cell population. They did in vivo transplantation experiments to show that CD34⁻CD133⁺GPI80⁺ cells (referred to as CD34⁻ SRCs/HSCs) could produce both CD34⁺ SRCs and CD34⁻ SRCs, while CD34⁺CD38⁻CD133⁺GPI80⁺ cells (referred to as CD34⁺ SRCs/HSCs) could only produce

CD34+ SRCs. Up to now, the most well-acceptable definition of human HSCs are Lin-CD34+CD38-CD45RA-CD90+CD49f+ cells (Sergei, et al., Cell Stem Cell, 2012; Faiyaz et al., Science, 2011; Iman et al., Science, 2014). Moreover, CD34+ HSCs have been shown to repopulate in a pre-clinical study (Wagner, et al., Cell Stem Cell, 2016). CD34- HSCs have only just been studied by few labs. If the authors wish to prove that Lin-CD34-CD133+ GPI-80+ HSCs are really the long-term HSC which localizes at the top of the human hematopoietic hierarchy, single cell multiple-generation transplantation should be performed to systematically compare their Lin-CD34-CD133+ GPI-80+ HSCs with the relatively well defined Lin-CD34+CD38-CD45RA-CD90+CD49f+ cell population.

Specific Comments (Major)

1. Be rigorous in your definitions. Are these cells truly CD34-? Does RT-PCR and Western analysis show lack of CD34 expression? Clarify why you don't just phenotype the cells as CD34-, rather than sometimes as CD34+/- or as CD34low/-. This is very confusing and makes it hard to evaluate your conclusions and interpretations.
2. What is the frequency of CD34- cells, CD34+ cells and Lin-CD34-CD133+GPI-80+ "HSCs" in mononuclear cells from single cord units?
3. CFU assay in most case is used as a functional assay for hematopoietic progenitor cells. Will HSC directly differentiate into progenitors to form colonies in the 10-14 days of the culture? Lin-CD34-CD133+GPI-80+ or Lin-CD34-CD133+GPI-80- cells tend to form CFU-EM. This may suggest that these cells are downstream progenitors with limited potential. Also, what's the in vivo situation with this phenotype? Based on this lineage bias in vitro, you cannot say the CD34- cells have this bias in vivo, as this bias is not seen in vivo.
4. According to the transplantation data, CD34-CD133+GPI80+ cells (referred to as CD34-SRCs/HSCs) generate both CD34+ SRCs and CD34- SRCs, while CD34+CD38-CD133+GPI80+ cells (referred to as CD34+ SRCs/HSCs) only produce CD34+ SRCs. What's the relation between Lin-CD34-CD133+GPI-80+ "HSCs" and Lin-CD34+CD38-CD45RA-CD90+CD49f+ HSCs? Single cell multiple-generation transplantation should be performed to determine Lin-CD34-CD133+ GPI-80+ HSCs or relatively well defined Lin-CD34+CD38-CD45RA-CD90+CD49f+. HSCs are the long-term reconstituting and self-renewing HSCs? Did the authors check the expression of CD90 and CD49f in these Lin-CD34-CD133+ GPI-80+ HSCs?
5. All transplantation data lack quantification, especially for the lineage staining such as for CD33, CD4 etc. both in primary transplantation and secondary transplantation? Most of this data is shown as flow plots. We need to see actual quantification for multiple mice and experiments.
6. Anjos-Afonso F et al., reported the presence of CD34- SRCs in Lin-CD34-CD38-CD93hi populations. Did the authors check CD93 expression levels in their Lin-CD34-CD133+GPI-80+ "HSCs"? Can the SRCs be further enriched when combined with CD93?
7. Did the authors check the expression of CXCR4, which is involved in homing and also in vivo maintenance of long-term HSCs? Surface expression of CXCR4 is dynamic, CD34+/CXCR4- sorted cells store intracellular CXCR4, which is rapidly induced to membrane expression within a few hours. Again, did the authors check the total expression of CD34 by qRT-PCR and western blot (involving both surface and intracellular CD34)?
8. When CD34+ CB HSC cells are cultured ex vivo, eventually CD34+ cells will become

CD34-. Can Lin-CD34-CD133+GPI-80+ "HSCs" be ex vivo expanded and give rise to more of these same cells and/or to CD34+ cells? You can use SCF, TPO, Flt3 ligand minus and plus SR1 or UM171 to test for expansion.

9. Signaling pathways like Notch, Wnt or TGFbeta are important for stem cell function and maintenance. The authors should check out these pathways in their CD34- HSC population.

10. What was the criteria the authors utilized to determine positively engrafted NSG mice? > 0.01% donor human CD45+ cells or a higher value? This would affect the Poisson statistical analysis of the limiting dilution assay and should be clarified.

Other Comments

1. Page 5, Second paragraph: Are the plating efficiencies statistically lower in the CD34- population?

2. Page 6, Bottom paragraph: How can you say: "...reached a peak level of repopulation at 12 weeks after transplantation." as no time points prior to 12 weeks are shown?

3. Page 7, Top 3 lines: What was the actual number of cells that engrafted?

4. Page 8, last line: How can you make a conclusion based on such a small n value?

5. Page 11, line 9 of first paragraph: Fix this sentence. Remove the words: The current

6. Page 12, lines 3-end of that paragraph: More discussion is needed on what you found may mean.

7. Page 12, 8 lines from bottom: close, not closed

8. Page 13, 8 lines from top: As noted above the cells may not be going directly to MEP bias, but may be biased to the in vitro culture condition.

9. Page 20, end of bottom paragraph: How does removing bone marrow aspirate effect later read-outs? Could this induce a state of "emerging hematopoiesis"? Does collecting aspirate vs. crushing bone at the end of the experiment lead to collection of different cell populations?

Reviewer #2 (Remarks to the Author):

Sumide et al. developed a method for purifying CD34- SRCs using two positive markers, CD133 and GPI-80. Using in vitro and in vivo functional assays, the authors showed that CD34+/- hematopoietic stem cells (HSCs) possess different lineage potentials. Using single-cell gene expression assays, the authors found both similarities and differences between the two types of HSCs. The authors proposed that megakaryocyte/erythrocyte progenitors are generated directly from CD34- HSCs (bypass route). They also proposed a revised road map for the commitment of human CD34- HSCs residing at the apex of the human HSC hierarchy. The study is of potential interests; however the following issues must be resolved before publication on Nature Communication.

1. When comparing the function of CD34- HSCs with CD34+ HSCs, the authors used their own marker system for the purification of CD34+ HSCs (18Lin-CD34+CD38-CD133+GPI-80+ cells). Analysis for a more established marker system should be also used for better comparison (F. Notta, S. Doulatov, E. Laurenti, A. Poepl, I. Jurisica, J. E. Dick, Isolation of single human hematopoietic stem cells capable of long-term multilineage engraftment,

Science, 333, 218–221 (2011))

2. The number of CD34⁻ HSCs seems to be much smaller than the CD34⁺ HSCs in the CB. How about bone marrow sample? What is significance of this HSC type if its frequency is low?
3. The organization of the abstract is poor. The sentences should be polished by a native speaker.
4. Fig 1i: The CD34⁻HSCs are more potent for MegE lineages. They are less potent to generate CFU-Mix. Maybe CD34⁻HSC is only a MegE progenitor, but stimulated for stem cell property in the irradiated mice?
5. Fig 2b: The Y axis is log scale, if transformed to normal scale the difference between CD34⁺ HSC and CD34⁻ HSC is huge (up to 50 fold). What is the explanation for that?
6. The single cell qPCR analysis part needs to be improved:
Fig 4c: MegE genes such as Gata1 and Klf1 should be shown. If shown, CD34⁻HSC might be clustered with MEP? Why some of the CD34⁻HSCs expressed CD34? Some of the CD34⁻HSCs do not show Actb expression?
Fig5a: Gata1 is expressed in CD34⁻ HSCs?
I suggest that more single cell qPCR assays with a broader panel of markers should be performed.
7. The transplantation results appear to be strong, but the conclusion that CD34⁻ HSCs reside at the apex of the human HSC hierarchy is not very convincing. The reconstitution ability of CD34⁻HSC appears to be much weaker than CD34⁺ HSC.
8. One thing to consider is that the marker system for human HSC is already complicated; the introduction of CD34⁻ HSCs using a brand new marker combination may further confuse the field. Perhaps, eventually, unbiased high throughput single cell sequencing of the CB or BM sample would help to define the true HSC cell type cluster with a better marker system.

Reviewer #3 (Remarks to the Author):

In this manuscript authors assess the engraftment and CFC potential of CD34⁻ CB HSCs. The studies are very meticulous and with high quality experiments. To precisely characterize CB CD34⁻ HSCs, they performed limiting dilution assay and single cell engraftment assay followed by secondary bone marrow transplantation. In addition, they revealed that CD34⁻ CB HSCs are highly enriched in erythro-megakaryocytic (E-Mk) CFCs.

Unfortunately, E-Mk potential of CD34⁻ cells is mostly addressed in vitro. In vivo studies, shows that CD34⁻ SRCs produce significantly less red blood cells compared CD34⁺ SRCs (Fig. 3). This finding is in contrast with the proposed hypothesis that CD34⁻ cells are an

important bypass route for E-Mk cells in vivo.

It would be important to assess erythroid output using a more sensitive system such as ex vivo culture of bone marrow xenograft (Hayakawa, Cell Transplant, 2010), NOD/SCID mice pretreated with clodronate or holo-transferrin, or by using NSGW41 mice or mice expressing human IL3 and EPO. If CD34⁻ cells do not possess E-Mk potential in vivo, it would indicate that CD34⁻ population include E-Mk CFCs without in vivo repopulation potential.

Authors should define the threshold for positive engraftment. Authors claim that CD34⁻ cells resorted from primary recipient show multilineage engraftment in secondary recipient. However, chimerism level obtained in secondary recipient is 0.01-0.02% (Table S2). This is an extremely low level (much lower than for CD34⁺ cells). I am not sure if this level is reliable and whether multilineage analysis at such a low level of engraftment is valid. Thus, the conclusion made by the authors that CD34⁻ HSCs cells are able to self-renew remains unproven. It is possible, that CD34⁻ cells have a limited self-renewal potential and represent a distinct type of HSCs with limited self-renewal potential, rather than HSCs that reside at the apex of human HSC hierarchy as depicted in Fig. 6b.

In engraftment analysis, a sizable proportion of hCD45⁺ cells are positive for mCD45. Why are hCD45⁺mCD45⁺ cells included in the analysis and why do the authors think that these are human cells and not mouse cells?

On page 9 "To provide an independent line of evidence for characterizing our highly purified CD34⁺/⁻ SRCs (HSCs), we analyzed gene expression profiles of these two classes of SRCs (HSCs) at the single-cell level." They should add that other types of progenitors and mature cells were analyzed as well. Otherwise, description of cluster analysis becomes confusing.

RESPONSE TO REVIEWER 1.

We thank the reviewer for these critical but very constructive comments on our manuscript.

1. *Be rigorous in your definitions. Are these cells truly CD34⁻? Does RT-PCR and Western analysis show lack of CD34 expression? Clarify why you don't just phenotype the cells as CD34⁻, rather than sometimes as CD34^{+/-} or as CD34^{low/-}. This is very confusing and makes it hard to evaluate your conclusions and interpretations.*

Response: We defined CD34⁻ cells by FACS using an FMO control, as shown in Appendix Fig. 1 and have now mentioned them on page 22, lines 9 to 11. CD34⁻ indicates an FACS-negative cell. As shown in revised Fig. 5c(ii) (original Fig. 5c(ii)), these CD34⁻ cells include both CD34 mRNA-positive and CD34 mRNA-negative cells. In addition, we confirmed that the expression of CD34 of FACS-sorted CD34⁻ cells was significantly lower than that of FACS-sorted CD34⁺ cells by a microarray analysis. However, it is very difficult to collect enough 18Lin⁻CD34⁻CD133⁺GPI-80⁺ cells to perform a Western blotting analysis because of their low incidence in the CB. As the reviewer pointed out, we used “CD34^{+/-} cells” many times in the original text. “CD34^{+/-} cells” mean CD34⁺ and CD34⁻ cells. We used this term simply because we want to reduce the length of our text to fit the provisions of Nature Medicine, the journal to which we first submitted this manuscript. We have therefore used CD34⁺ and CD34⁻ instead of CD34^{+/-} throughout the text and supplementary information in this revised version. On page 3, lines 11 to 12 in the original text, we used CD34^{low/-} KSL cells, which indicates that these cells contained CD34^{low} and CD34⁻ cell populations.

Appendix Figure 1.

2. *What is the frequency of CD34⁻ cells, CD34⁺ cells and Lin-CD34-CD133⁺GPI-80⁺ “HSCs” in mononuclear cells from single cord units?*

Response: Empirically, there is a wide distribution in the numbers of CD34⁻, CD34⁺ and 18Lin⁻CD34⁻CD133⁺GPI-80⁺ “HSCs” in single cord blood units. We therefore calculated the frequency of these cells among total nucleated cells (TNCs) instead of mononuclear cells (MNCs), as we did not use Ficoll-Paque to collect MNCs in our present study. As shown in Appendix Table 1, the mean frequencies of 18Lin⁻CD34⁺ and 18Lin⁻CD34⁻ cells are 1/2,255 and 1/32,203, respectively. In addition, the mean frequencies of 18Lin⁻CD34⁺CD38⁻CD133⁺GPI-80⁺ and 18Lin⁻CD34⁻CD133⁺GPI-80⁺ cells are 1/282,582 and 1/5,253,326, respectively.

Appendix Table 1.

n=32	Frequency of the target cells in TNCs			
	18Lin ⁻ CD34 ⁺ cells	18Lin ⁻ CD34 ⁻ cells	34 ⁺ 38 ⁻ 133 ⁺ 80 ⁺ SRCs	34 ⁻ 133 ⁺ 80 ⁺ SRCs
Mean	1/2,255	1/32,203	1/282,582	1/5,253,326
Median	1/3,243	1/54,783	1/404,179	1/7,074,571
Range	1/31,411 - 1/604	1/951,510 - 1/8,350	1/2,453,177 - 1/75,170	1/122,250,000 - 1/1,750,000

TNC = Total nucleated cell residing in one CB unit.

3. *CFU assay in most case is used as a functional assay for hematopoietic progenitor cells. Will HSC directly differentiate into progenitors to form colonies in the 10-14 days of the culture? Lin-CD34-CD133+GPI-80+ or Lin-CD34-CD133+GPI-80- cells tend to form CFU-EM. This may suggests that these cells are downstream progenitors with limited potential. Also, what's the in vivo situation with this phenotype? Based on this lineage bias in vitro, you cannot say the CD34- cells have this bias in vivo, as this bias is not seen in vivo.*

Response: This is very important critique. In the CFC assay shown in revised Figs.1i and j (original Figs.1i and j) and revised Fig. S2 (original Fig. S1), we cultured FACS-sorted single 18Lin⁻CD34⁺CD38⁻CD133⁺GPI-80⁺ or 18Lin⁻CD34⁻CD133⁺GPI-80⁻ cells in methylcellulose using the ACUDU. Both of the cells contain CD34⁺ (incidence, 1/5) and CD34⁻ (incidence, 1/8) SRCs (HSCs) as well as hematopoietic progenitor cells (HPCs). Therefore, we believe that both HSCs and HPCs can form various types of hematopoietic colonies as shown in revised Fig. S2 (original Fig. S1). Unexpectedly, 18Lin⁻CD34⁻CD133⁺GPI-80⁺ or 18Lin⁻CD34⁻CD133⁺GPI-80⁻ cells tended to form CFU-EM. In order to confirm the megakaryocyte/erythrocyte lineage bias *in vivo*, we performed additional analyses of multilineage differentiation potentials of CD34⁺ and CD34⁻ SRCs *in vivo* using the SRC assay data shown in original Fig. 2 (revised Fig. 2) and original Table S2 (revised Table S2). As shown in revised Fig. S8a (new data), CD34⁻ SRCs showed a significantly higher megakaryocyte/erythrocyte lineage differentiation potential *in vivo*, as we observed *in vitro*. The reviewer suggests that 18Lin⁻CD34⁻CD133⁺GPI-80⁺ cells are downstream progenitors with limited potential. However, as shown in revised Fig. S12 (original Fig. S6), the gene expression profiles of

these cells are clearly different from that of MEPs. Furthermore, when we transplanted these cells as single cells into NSG mice, they were able to repopulate multi-lineage human hematopoiesis in primary, secondary and tertiary recipient mice for over one year, as shown in revised Table S3 (original Table S3), revised Fig. 3 (original Fig. 3), and revised Figs. S9a to d (original Fig. S5a to c). These results indicate that CD34⁻ SRCs are multi-potential long-term repopulating HSCs. We have now described these findings on page 8, line 26 to page 9, line 15, page 10, line 16 to page 11, line 22 and on page 19, lines 10 to 15.

4. *According to the transplantation data, CD34⁻CD133⁺GPI80⁺ cells (referred to as CD34⁻ SRCs/HSCs) generate both CD34⁺ SRCs and CD34⁻ SRCs, while CD34⁺CD38⁻CD133⁺GPI80⁺ cells (referred to as CD34⁺ SRCs/HSCs) only produce CD34⁺ SRCs. What's the relation between Lin⁻CD34⁻CD133⁺GPI-80⁺ "HSCs" and Lin⁻CD34⁺CD38⁻CD45RA⁻CD90⁺CD49f⁺ HSCs? Single cell multiple-generation transplantation should be performed to determine Lin⁻CD34⁻CD133⁺ GPI-80⁺ HSCs or relatively well defined Lin⁻CD34⁺CD38⁻CD45RA⁻CD90⁺CD49f⁺. HSCs are the long-term reconstituting and self-renewing HSCs? Did the authors check the expression of CD90 and CD49f in these Lin⁻CD34⁻CD133⁺ GPI-80⁺ HSCs?*

Response: As suggested, we performed additional experiments to analyze the expression of CD90 and CD49f on the surfaces of 18Lin⁻CD34⁺CD38⁻CD133⁺GPI-80⁺ and 18Lin⁻CD34⁻CD133⁺GPI-80⁺ cells by FCM and presented these data in revised Fig. S1 (new data). As shown in this figure, most of the 18Lin⁻CD34⁺CD38⁻CD133⁺GPI-80⁺ and 18Lin⁻CD34⁻CD133⁺GPI-80⁺ cells expressed CD90 and CD49f. Thus, our highly purified 18Lin⁻CD34⁺CD38⁻CD133⁺GPI-80⁺ cells (HSCs) overlapped with 9Lin⁻CD34⁺CD38⁻CD45RA⁻CD90⁺CD49f⁺ cells (HSCs). However, the incidence of SRCs (HSCs) in 9Lin⁻CD34⁺CD38⁻CD45RA⁻CD90⁺CD49f⁺ cells was 1/10.5 (Science 333:218-221,2011). In contrast, the incidence of SRCs (HSCs) in 18Lin⁻CD34⁺CD38⁻CD133⁺GPI-80⁺ cells was 1/5, as shown in revised Fig. S5 (original Fig. S3). These results demonstrated that CD34⁺ HSCs are more enriched in our purified 18Lin⁻CD34⁺CD38⁻CD133⁺GPI-80⁺ cells than 9Lin⁻CD34⁺CD38⁻CD45RA⁻CD90⁺CD49f⁺ cells. In this revised version, we added data of single-cell initiated tertiary transplantation. As shown in revised Table S3 (original Table S3), revised Fig. 3 (original Fig. 3), and revised Figs. S9a to d (original Fig. S5a to c), single 18Lin⁻CD34⁺CD38⁻CD133⁺GPI-80⁺ as well as 18Lin⁻CD34⁻CD133⁺GPI-80⁺ cells are able to repopulate multi-lineage human hematopoiesis in primary, secondary and tertiary recipient mice for over one year. These results indicate that 18Lin⁻CD34⁺CD38⁻CD133⁺GPI-80⁺ as well as 18Lin⁻CD34⁻CD133⁺GPI-80⁺ HSCs are long-term repopulating and self-renewing HSCs. We have now described these findings on page 5, lines 13 to 18, on page 11, lines 3 to 22,

and on page 16, lines 10 to 20.

5. *All transplantation data lack quantification, especially for the lineage staining such as for CD33, CD4 etc. both in primary transplantation and secondary transplantation? Most of this data is shown as flow plots. We need to see actual quantification for multiple mice and experiments.*

Response: As suggested, we performed additional quantitative analyses of multi-lineage differentiation potentials of CD34⁺ and CD34⁻ SRCs using the SRC assay data shown in original Fig. 2 (revised Fig. 2) and original Table S2 (revised Table S2). As shown in revised Fig. S8a (new data), the human CD45⁺ cell repopulation capacities of both of the CD34⁺ and CD34⁻ SRCs were not significantly different. In addition, both CD34⁺ and CD34⁻ SRCs showed multi-lineage differentiation potentials, including CD34, CD33, CD19, CD14, CD11b, CD41, CD235a, CD3, CD4 and CD8, in the BM, PB, spleen and thymus in primary transplantation. The numbers of mice analyzed are denoted in the bottom of each figure. These quantitative analyses clearly demonstrated that CD34⁻ SRCs showed a significantly higher megakaryocyte/erythrocyte differentiation potential than CD34⁺ SRCs. We also analyzed multi-lineage differentiation potentials of CD34⁺ and CD34⁻ SRCs in the BM, PB and spleen in secondary transplantation, as shown in revised Fig. S8b (new data). The human CD45⁺ cell repopulation capacities of both CD34⁺ and CD34⁻ SRCs were not significantly different. The numbers of mice analyzed are denoted in the bottom of each figure. However, these CD34⁻ SRCs showed significantly higher levels of CD34⁺ cells in both of the tibiae (injected site) and other bones than CD34⁺ SRCs. These results suggest that CD34⁻ SRCs are a more immature class of HSCs than CD34⁺ SRCs. We have now described these findings on page 8, line 26 to page 9, line 22, and on page 19, line 22 to page 20, line 3.

6. *Anjos-Afonso F et al., reported the presence of CD34⁻ SRCs in Lin⁻CD34⁻CD38⁻CD93^{hi} populations. Did the authors check CD93 expression levels in their Lin⁻CD34⁻CD133⁺GPI-80⁺ “HSCs”? Can the SRCs be further enriched when combined with CD93?*

Response: As suggested, we performed additional experiments to analyze the expression of CD93 on the surfaces of 18Lin⁻CD34⁺CD38⁻CD133⁺GPI-80⁺ and 18Lin⁻CD34⁻CD133⁺GPI-80⁺ cells by FCM and presented these data as revised Fig. S1 (new data). As clearly shown in this figure, approximately 40 % of 18Lin⁻CD34⁺CD38⁻CD133⁺GPI-80⁺ cells expressed CD93. In contrast, none of 18Lin⁻CD34⁻CD133⁺GPI-80⁺ cells expressed CD93. Thus, CD93 cannot be used for the further enrichment of CD34⁻ SRCs. As shown in revised Fig. S11 (original Fig. S9), CD93

was detected as one of the genes enriched in non-SRCs. These results are consistent with our previous report (Leukemia 28:1308-1315,2014). We have now described these findings on page 5, lines 13 to 18, and on page 15, lines 1 to 4.

7. *Did the authors check the expression of CXCR4, which is involved in homing and also in vivo maintenance of long-term HSCs? Surface expression of CXCR4 is dynamic, CD34+/CXCR4- sorted cells store intracellular CXCR4, which is rapidly induced to membrane expression within a few hours. Again, did the authors check the total expression of CD34 by qRT-PCR and western blot (involving both surface and intracellular CD34)*

Response: As suggested, we performed additional experiments to analyze the expression of CXCR4 on the surfaces of 18Lin⁻CD34⁺CD38⁻CD133⁺GPI-80⁺ and 18Lin⁻CD34⁻CD133⁺GPI-80⁺ cells by FCM and presented these data in revised Fig. S1 (new data). As shown in this figure, most of 18Lin⁻CD34⁺CD38⁻CD133⁺GPI-80⁺ and 18Lin⁻CD34⁻CD133⁺GPI-80⁺ cells expressed CXCR4. We assessed the expression of CD34 via single-cell qRT-PCR as shown in revised Fig. 5c(ii) (original Fig. 5c(ii)). However, it is very difficult to collect enough 18Lin⁻CD34⁻CD133⁺GPI-80⁺ cells to perform a Western blotting analysis because of their low incidence in the CB, as shown in Appendix Table 1. We have now described these findings on page 5, lines 13 to 16.

Appendix Table 1.

n=32	Frequency of the target cells in TNCs			
	18Lin ⁻ CD34 ⁺ cells	18Lin ⁻ CD34 ⁻ cells	34 ⁺ 38 ⁻ 133 ⁺ 80 ⁺ SRCs	34 ⁻ 133 ⁺ 80 ⁺ SRCs
Mean	1/2,255	1/32,203	1/282,582	1/5,253,326
Median	1/3,243	1/54,783	1/404,179	1/7,074,571
Range	1/31,411 - 1/604	1/951,510 - 1/8,350	1/2,453,177 - 1/75,170	1/122,250,000 - 1/1,750,000

TNC = Total nucleated cell residing in one CB unit.

8. *When CD34+ CB HSC cells are cultured ex vivo, eventually CD34+ cells will become CD34-. Can Lin-CD34-CD133+GPI-80+ “HSCs” be ex vivo expanded and give rise to more of these same cells and/or to CD34+ cells? You can use SCF, TPO, Flt3 ligand minus and plus SR1 or UM171 to test for expansion.*

Response: We are very interested in the *ex vivo* expansion of CD34⁺ and CD34⁻ SRCs (HSCs) using SCF, TPO, Flt3 ligand minus and plus SR1 or UM171. We plan to perform such experiments in the next study.

9. *Signaling pathways like Notch, Wnt or TGFbeta are important for stem cell function and maintenance. The authors should check out these pathways in their CD34- HSC population.*

Response: As suggested, we performed additional analyses of signaling pathways like Notch, Wnt or TGF- β using the microarray data. As shown in revised Fig. 6 (original Fig. S8), we focused on the signaling pathways important for the HSC function and maintenance. Interestingly, CD34⁻ HSCs expressed higher levels of gene sets related to Wnt signaling (revised Figure 6b), while CD34⁺ HSCs expressed higher levels of gene sets related to IFN α , IFN γ , TGF- β and Notch signaling (revised Figure 6c). We have now described these findings on page 13, lines 19 to 22.

10. *What was the criteria the authors utilized to determine positively engrafted NSG mice? > 0.01% donor human CD45⁺ cells or a higher value? This would affect the Poisson statistical analysis of the limiting dilution assay and should be clarified.*

Response: As previously reported (Leukemia 28:1308-1315,2014), the mice were scored as positive if more than 0.01% of the total murine BM cells were human CD45⁺ cells. In separate experiments, we confirmed that the detection limit of human CD45⁺ cells in mouse BMs by an FCM analysis was 0.005%. Please refer to the data shown in Fig. S1 which was published previously (Leukemia 28:1308,2014). In our LDA analysis shown in revised Fig. S5 (original Fig. S3), the actual repopulation rates of human CD45⁺ cells ranged from 0.06 % to 69.8 %. In this revised version, we presented new data in revised Figs. S7c and S9d (upper column). In these figures, the repopulation rates of human CD45⁺ cells in these two mice were 0.02 %. Both mice clearly showed lympho-myeloid repopulation. We performed the LDA based on these accurate analyses. Thus, our criteria for positive engraftment did not affect the Poisson statistical analysis of the LDA. We have presented the LDA data on page 7, lines 13 to 21. We have also added the actual repopulation rates of human CD45⁺ cells in the LDA to the figure legend for revised Fig. S5.

Other Comments:

1. *Page 5, Second paragraph: Are the plating efficiencies statistically lower in the CD34-population?*

Response: As suggested, we performed a statistical analysis using a two-way analysis of variance. As shown in revised Fig. 1i (original Fig. 1i), the plating efficiencies (PEs) of 34⁺38⁻133⁺80⁻ and 34⁻133⁺80⁻ cells were significantly higher than those of 34⁺38⁻133⁺80⁺ and 34⁻133⁺80⁺ cells. Furthermore, the PEs of 34⁺38⁻133⁺ cells were also significantly higher than those of 34⁻133⁺ cells regardless of the GPI-80 expression. We described these findings on page 5, lines 24 to 27. In addition, we also described the statistical analysis methods in page 34, lines 9 to 11.

2. *Page 6, Bottom paragraph: How can you say: "...reached a peak level of repopulation at 12 weeks after transplantation." as no time points prior to 12 weeks are shown?*

Response: In response to this comment, we changed our description from "All of the primary recipient mice showed signs of human cell repopulation and reached a peak level of repopulation at 12 weeks after transplantation" to "All of the primary recipient mice showed signs of human cell repopulation at 12 weeks after transplantation". We have now described these findings on page 7, line 27 to page 8, line 2.

3. *Page 7, Top 3 lines: What was the actual number of cells that engrafted?*

Response: In the original Table S2, we presented the numbers of whole BM cells containing murine and human cells. As suggested, we calculated the actual numbers of human cells transplanted into the secondary recipient mice and presented these data in revised Table S2.

4. *Page 8, last line: How can you make a conclusion based on such a small n value?*

Response: We understand the reviewer's concern here. On page 8, last line in the original text, we described the data of one representative NSG mouse (34⁻ NSG058). However, as shown in revised Table S3 (original Table S3), one NSG mouse (CD34⁺046) and another four NSG mice (34⁻ NSG027, NSG058, NSG076 and NSG079) also showed distinct multi-lineage human cell repopulation in two secondary recipient mice by 21 to 23 weeks after transplantation. Based on the repopulation data obtained from a total of 10 secondary recipient mice, we suggested that daughter SRCs generated from the single parental CD34⁺ and CD34⁻ SRCs were functionally heterogeneous. We have now described these findings on page 10, line 26 to page 11, line 11.

5. *Page 11, line 9 of first paragraph: Fix this sentence. Remove the words: The current*

Response: As suggested, we remove the words "The current" from page 11, line 9 in the original text. The revised sentence "Our updated method enables us to perform single-cell-based transplantation analyses" appears on page 16, lines 10 to 11 in this revised version.

6. *Page 12, lines 3-end of that paragraph: More discussion is needed on what you found may mean.*

Response: As suggested, we added more discussion of what we found by single-cell qRT-PCR and microarray analyses. In the original text, we mainly described the global gene expression analysis of CD34⁺ and CD34⁻ HSCs in the Supplementary information. In the revised version, we describe these findings in the Result section on page 13, line 4 to page 15, line 6. In the Discussion, we also mentioned the significance of our gene expression analyses on page 17, lines 6 to 13, and on page 17, line 16 to page 18, line 3. Based on these changes, we also revised Fig. 6 (original Fig. S8) .

7. *Page 12, 8 lines from bottom: close, not closed*

Response: As suggested, we corrected the mistake on page 18, line 13, in this revised version.

8. *Page 13, 8 lines from top: As noted above the cells may not be going directly to MEP bias, but may be biased to the in vitro culture condition.*

Response: In the previous version, we did not quantitatively analyze the multi-lineage repopulation potentials of CD34⁺ and CD34⁻ SRCs (HSCs). As suggested, we performed additional quantitative analyses of the multi-lineage differentiation potentials of CD34⁺ and CD34⁻ SRCs using the SRC assay data shown in original Fig. 2 and original Table S2. As shown in revised Fig. 8a (new data), CD34⁻ SRCs expressed significantly higher megakaryocyte/erythrocyte differentiation potentials than CD34⁺ SRCs *in vivo*. This suggests that CD34⁻ HSCs may directly differentiate into MEPs *in vitro* as well as *in vivo*. We have now described these findings on page 8, line 26 to page 9, line 15, and on page 19, line 24 to page 20, line 3.

9. *Page 20, end of bottom paragraph: How does removing bone marrow aspirate effect later read-outs? Could this induce a state of “emerging hematopoiesis”? Does collecting aspirate vs. crushing bone at the end of the experiment lead to collection of different cell populations?*

Response: In order to address the reviewer’s comment, we performed an additional experiment and compared the findings from two repopulation analyses in the same mouse. First, we analyzed the human hematopoietic cell repopulation at 18 weeks after transplantation by the BM aspiration method. We then euthanized the mouse at 20 weeks after transplantation and analyzed the human hematopoietic cell repopulation using the bone crushing method. As shown in Appendix Figs. 2a and b, the two repopulation patterns were comparable. Based on these data, we believe that the BM aspiration method does not influence later read-outs.

Appendix Figure 2

RESPONSE TO REVIEWER 2.

We thank the reviewer for these critical but very constructive comments on our manuscript.

1. *When comparing the function of CD34⁻ HSCs with CD34⁺ HSCs, the authors used their own marker system for the purification of CD34⁺ HSCs (18Lin⁻CD34⁺CD38⁻CD133⁺GPI-80⁺ cells). Analysis for a more established marker system should be also used for better comparison (F. Notta, S. Doulatov, E. Laurenti, A. Poeppel, I. Jurisica, J. E. Dick, Isolation of single human hematopoietic stem cells capable of long-term multilineage engraftment, *Science*, 333, 218–221 (2011))*

Response: As suggested, we performed additional experiments to analyze the expression of CD90 and CD49f on the surfaces of 18Lin⁻CD34⁺CD38⁻CD133⁺GPI-80⁺ and 18Lin⁻CD34⁻CD133⁺GPI-80⁺ cells by FCM and presented these data in revised Fig. S1 (new data). As shown in this figure, most of the 18Lin⁻CD34⁺CD38⁻CD133⁺GPI-80⁺ and 18Lin⁻CD34⁻CD133⁺GPI-80⁺ cells expressed CD90 and CD49f. Thus, our highly purified 18Lin⁻CD34⁺CD38⁻CD133⁺GPI-80⁺ cells (HSCs) overlapped with 9Lin⁻CD34⁺CD38⁻CD45RA⁻CD90⁺CD49f⁺ cells (HSCs). However, the incidence of SRCs (HSCs) in 9Lin⁻CD34⁺CD38⁻CD45RA⁻CD90⁺CD49f⁺ cells was 1/10.5 (*Science* 333:218-221,2011). In contrast, the incidence of SRCs (HSCs) in 18Lin⁻CD34⁺CD38⁻CD133⁺GPI-80⁺ cells was 1/5, as shown in revised Fig. S5 (original Fig. S3). These results demonstrated that CD34⁺ HSCs are more enriched in our purified 18Lin⁻CD34⁺CD38⁻CD133⁺GPI-80⁺ cells than 9Lin⁻CD34⁺CD38⁻CD45RA⁻CD90⁺CD49f⁺ cells. We have now described these findings on page 5, lines 13 to 16, and on page 7, lines 13 to 18.

2. *The number of CD34⁻ HSCs seems to be much smaller than the CD34⁺ HSCs in the CB. How about bone marrow sample? What is significance of this HSC type if its frequency is low?*

As pointed out by the reviewer, the number of CD34⁻ HSCs was much smaller than that of CD34⁺ HSCs in the CB. As shown in Appendix Table 1, the mean frequencies of 18Lin⁻CD34⁺CD38⁻CD133⁺GPI-80⁺ cells (CD34⁺ HSCs) and 18Lin⁻CD34⁻CD133⁺GPI-80⁺ cells (CD34⁻ HSCs) among total nucleated cells were 1/282,582 and 1/5,253,326, respectively. We also investigated the presence of 18Lin⁻CD34^{+/+}CD133⁺ cells in the adult human BM cells. As shown in Appendix Fig. 3, approximately 60 % of 18Lin⁻CD34⁺ cells are CD133⁺. In contrast, it is hard to detect 18Lin⁻CD34⁻CD133⁺ cells in adult human BM. As we previously reported (*Leukemia* 28:1308,2014), all of the CB-derived CD34⁺ and CD34⁻ SRCs were detected in

18Lin⁻CD34^{+/}-CD133⁺ cell fractions. Thus, adult BM cells may only contain CD34⁺ HSCs. Further analyses will be needed to clarify this important issue.

As mentioned above, the number of CD34⁻ HSCs in the CB is much smaller than that of the CD34⁺ HSCs. However, as shown in revised Figs. 4 to 6 (original Figs. 4, 5 and S8), CD34⁻ and CD34⁺ HSCs showed different gene expression profiles. Furthermore, CD34⁻ HSCs showed a potent megakaryocyte/erythrocyte differentiation potential *in vitro* and *in vivo* (revised Figs. 1i and j, and Fig. S8a [new data]). These results suggest that CD34⁻ HSCs are different class of HSCs from CD34⁺ HSCs. In addition, as shown in revised Fig. S10 (new data), CD34⁺ and CD34⁻ HSCs expressed comparable levels of *IFNR1* and *R2* receptors, *TYK2* and *JAK1*. However, CD34⁻ HSCs expressed significantly lower levels of their downstream target genes, including *STAT1*, *IFITM1*, *IFITM3*, *DDX58*, *IFI44*, *CXCL10* and *CXCL11*, than CD34⁺ HSCs. As recently reported, high levels of IFN α activated dormant HSCs *in vivo* (Nature 458:904,2009; Nat Med 15:696,2009). Thus, these results suggest that CD34⁻ HSCs may be more resistant than CD34⁺ HSCs to chronic activation of the IFN α pathway. These unique characteristics of CD34⁻ HSCs may play an important role in the emerging hematopoiesis. We have now described these findings on page 13, line 24 to page 14, line 6, and on page 17, line 17 to page 18, line 3.

Appendix Table 1.

n=32	Frequency of the target cells in TNCs			
	18Lin ⁻ CD34 ⁺ cells	18Lin ⁻ CD34 ⁻ cells	34 ⁺ 38 ⁻ 133 ⁺ 80 ⁺ SRCs	34 ⁻ 133 ⁺ 80 ⁺ SRCs
Mean	1/2,255	1/32,203	1/282,582	1/5,253,326
Median	1/3,243	1/54,783	1/404,179	1/7,074,571
Range	1/31,411 - 1/604	1/951,510 - 1/8,350	1/2,453,177 - 1/75,170	1/122,250,000 - 1/1,750,000

TNC = Total nucleated cell residing in one CB unit.

Appendix Figure 3

3. *The organization of the abstract is poor. The sentences should be polished by a native speaker.*

Response: According to the reviewer's comment, we revised the abstract and checked by a native speaker.

4. *Fig 1i: The CD34-HSCs are more potent for MegE lineages. They are less potent to generate CFU-Mix. Maybe CD34-HSC is only a MegE progenitor, but stimulated for stem cell property in the irradiated mice?*

Response: The reviewer suggests that the CD34⁻ HSCs are only a MegE progenitor. However, we respectfully object to this. First of all, in this study, we performed single-cell initiated serial transplantation analyses. As shown in revised Table S3 (original Table S3) and revised Figs. S9a to d (original Figs. S5a to c), single CD34⁻ SRCs (HSCs) engrafted primary, secondary and tertiary recipient mice with multi-lineage repopulation for over one year. These results clearly demonstrated that CD34⁻ SRCs are multi-potent long-term repopulating HSCs. In addition, we compared their gene expression profiles at the single-cell level. As shown in revised Fig. S12 (original Fig. S6), the gene expression profiles of CD34⁻ HSCs and MEPs are clearly different. These data demonstrated that the CD34⁻ HSCs are not MegE progenitors. We have now described these findings on page 11, lines 12 to 22, on page 16, lines 16 to 20, and on page 19, lines 10 to 15.

5. *Fig 2b: The Y axis is log scale, if transformed to normal scale the difference between CD34⁺ HSC and CD34⁻ HSC is huge (up to 50 fold). What is the explanation for that?*

Response: As pointed out by the reviewer, the repopulation levels of CD34⁺ SRCs are significantly higher than those of CD34⁻ SRCs at 12 and 18 weeks after transplantation (revised Fig. 2b) (original Fig. 2b). However, the repopulation levels of both of the CD34⁺ and CD34⁻ SRCs were comparable at 20 to 22 weeks after transplantation. We believe that one reason for this is the difference in the cell cycle status of CD34⁺ and CD34⁻ SRCs. As shown in revised Fig. S3a (original Fig. S2), approximately 50 % of CD34⁻ HSCs are in G₀ phase of the cell cycle. In contrast, only 30 % of CD34⁺ HSCs are in G₀ phase of the cell cycle. Thus, the repopulation level of CD34⁻ SRCs increase more slowly than that of CD34⁺ SRCs, as shown in revised Fig. 2b. We have now described these findings on page 6, lines 9 to 12.

In addition, we transplanted 200 18Lin⁻CD34⁺CD38⁻CD133⁺GPI-80⁺ cells (containing 41 CD34⁺ SRCs) and 200 18Lin⁻CD34⁻CD133⁺GPI-80⁺ cells (containing 25 CD34⁻ SRCs). This difference in the numbers of SRCs transplanted to each recipient mouse may also have affected the difference in the repopulation levels during the

observation period. We have now described these findings on page 7, line 25 to page 8, line 5.

Furthermore, in this revised version, we performed additional quantitative analyses of the repopulation rates of human CD45⁺ cells in the primary recipient mice that received 41 CD34⁺ SRCs and 25 CD34⁻ SRCs, as shown in revised Fig. S8a (new data). As clearly seen in this figure, there were no marked differences in the human CD45⁺ cell repopulation rates between CD34⁺ and CD34⁻ SRCs. We have now described these findings on page 8, line 27 to page 9, line 4.

6. *The single cell qPCR analysis part needs to be improved:*

Fig 4c: MegE genes such as Gata1 and Klf1 should be shown. If shown, CD34-HSC might be clustered with MEP? Why some of the CD34-HSCs expressed CD34? Some of the CD34-HSCs do not show Actb expression? Fig5a: Gata1 is expressed in CD34-HSCs? I suggest that more single cell qPCR assays with a broader panel of markers should be performed.

Response: We tried to detect *GATA1* and *KLF1* genes in a series of three single-cell qRT-PCR analyses. However, we were unable to detect these genes. We therefore believe that the expression levels of these genes may be low. Instead of these genes, we analyzed several genes related to MegE differentiation, including *NFE2*, *GFI1B*, *GATA2*, *TFRC*, *MYC* and *MYB*, at the single-cell level. As shown in revised Fig. S12 (original Fig. S6), the gene expression profiles of the above-mentioned MegE genes of CD34⁻ HSCs and MEPs were clearly different. We have now described these findings on page 19, lines 10 to 15.

We defined CD34⁻ cells by FACS using an FMO control as shown in Appendix figure 1. Therefore, CD34⁻ means a FACS-negative cell. As shown in revised Fig. 5c(ii) (original Fig. 5c(ii)), these CD34⁻ cells include both CD34 mRNA-positive and CD34 mRNA-negative cells. We have now described these findings on page 22, lines 9 to 11.

The reviewer mentioned that “some of the CD34⁻ HSCs did not show Actb expression”. However, we believe that the reviewer may have misunderstood our data. As shown in revised Fig. 5c(i) (original Fig. 5c(i)), all of the single cells analyzed expressed reference genes, including *ACTB*, *GAPDH*, *PTRC* and *PGK1*.

In the single-cell qRT-PCR analysis, we referenced a panel of genes reported in the literature (Cell 132:631,2008). As shown in revised Figs. 4 and 5 (original Figs. 4 and 5), we were able to detect clear differences in the gene expression profiles of CD34⁺ and CD34⁻ HSCs. Furthermore, the MegE gene expression profiles of CD34⁻ HSCs and MEPs were clearly different, as shown in Fig. S12. We therefore believe that the panel of genes used in this study, which are critical for the HSC maintenance and function, may be within the permissible range. We have now described these findings on page 11, line 24 to page

13, line 2.

Appendix Figure 1.

7. *The transplantation results appear to be strong, but the conclusion that CD34⁻ HSCs reside at the apex of the human HSC hierarchy is not very convincing. The reconstitution ability of CD34⁻ HSC appears to be much weaker than CD34⁺ HSC.*

Response: The reviewer pointed out that “The reconstitution ability of CD34⁻ HSC appears to be much weaker than CD34⁺ HSC.” However, we think that the reviewer may have misunderstood our data. In the first set of transplantation analyses (revised Fig. 2 (original Fig. 2)), we transplanted 200 18Lin⁻CD34⁺CD38⁻CD133⁺GPI-80⁺ cells (containing 41 CD34⁺ SRCs) and 200 18Lin⁻CD34⁻CD133⁺GPI-80⁺ cells (containing 25 CD34⁻ SRCs). Their repopulation kinetics differed slightly, as mentioned in our response to comment 5 of this reviewer. However, the repopulation levels of CD34⁺ and CD34⁻ SRCs were comparable at 20 to 22 weeks after transplantation. In other words, 25 CD34⁻ SRCs demonstrated an almost equivalent repopulation ability to 41 CD34⁺ SRCs. These results suggest that the reconstitution ability of CD34⁻ SRCs may be much higher than that of CD34⁺ SRCs. We have now described these findings on page 7, line 25 to page 8, line 5.

In the single-cell-initiated serial transplantation analysis, CD34⁻ SRCs engrafted primary, secondary and tertiary NSG recipient mice, just like CD34⁺ SRCs did (revised Table S3) (original Table S3). The incidences of engraftment of SRCs, such as CD34⁺ (5/6 [83.3 %]) and CD34⁻ (4/6 [66.7 %]) in the secondary recipient mice, were comparable. Notably, the incidences of engraftment of the CD34⁺ and CD34⁻ SRCs in the tertiary recipient mice are 1/3 (33.3%) for CD34⁺ SRCs, and 2/4 (50%) for CD34⁻ SRCs. Thus, the reconstitution ability of CD34⁻ SRCs in tertiary transplantation appeared to be comparable to that of CD34⁺ SRCs at the single-cell level. We have now described these findings on page 9, line 24 to page 11, line 22.

In our previous series of studies, we reported that CD34⁻ SRCs can generate CD34⁺ SRCs *in vitro* (Stem Cells 25:1348,2007; Stem Cells 33:1554,2015; Blood

128:2258,2016) and *in vivo* (Blood 101:2924,2003; Cell Transplant 26:1043,2017). However, CD34⁺ SRCs cannot generate CD34⁻ SRCs. In this study, as shown in revised Table S2 (original Table S2), resorted CD 34⁺ and CD34⁻ cells generated from a primary recipient mouse that received CD34⁻ SRCs were able to repopulate significant numbers of secondary recipient mice, as shown in Fig. S7 (new data). In contrast, resorted CD34⁻ cells generated from a primary recipient mouse that received CD34⁺ SRCs were unable to repopulate secondary recipient mice. Only CD34⁺ cells were able to repopulate significant numbers of secondary recipient mice. These results suggest that CD34⁻ SRCs (HSCs) reside at the apex of the human HSC hierarchy. We have now described these findings on page 8, lines 6 to 24.

8. *One thing to consider is that the marker system for human HSC is already complicated; the introduction of CD34- HSCs using a brand new marker combination may further confuse the field. Perhaps, eventually, unbiased high throughput single cell sequencing of the CB or BM sample would help to define the true HSC cell type cluster with a better marker system.*

Response: We agree with this reviewer and believe that unbiased high-throughput single-cell sequencing of the CB- or BM-derived HSC population would help to define the true HSC phenotype. However, it would then be very important to use highly purified target cells. To achieve this, we must purify the target cell population as much as possible using known marker systems as well as our newly reported markers. We hope that our present findings will contribute to this field of research, even if only slightly.

RESPONSE TO REVIEWER 3.

We thank the reviewer for these critical but very constructive comments on our manuscript.

1. *Unfortunately, E-Mk potential of CD34⁻ cells is mostly addressed in vitro. In vivo studies, shows that CD34⁻ SRCs produce significantly less red blood cells compared CD34⁺ SRCs (Fig. 3). This finding is in contrast with the proposed hypothesis that CD34⁻ cells are an important bypass route for E-Mk cells in vivo.*

Response: As the reviewer pointed out, in these representative mice that received CD34⁺ and CD34⁻ SRCs shown in original Figs. 3a and b (revised Figs. 3a and b), the proportion of CD235a⁺ cells produced by CD34⁻ SRCs appears to be lower than that produced by CD34⁺ SRCs. However, as shown in revised Figs. S9b and c (original Figs. S5b and c), CD235a⁺ erythroid cells were detectable only in the secondary recipient mice (34⁻ NSG 058a and b) that received BM cells from primary recipient mouse engrafted by CD34⁻ SRCs. We have now described these findings on page 10, line 27 to page 11, line 11.

There therefore seems to be a substantial individual difference between CD34⁺ and CD34⁻ SRCs (HSCs). These findings may reflect the heterogeneity of gene expression between the two, as shown in revised Figs. 4 to 6 (original Figs. 4, 5 and S8).

In order to address this reviewer's comment, we performed additional quantitative analyses of multi-lineage differentiation potentials of CD34⁺ and CD34⁻ SRCs using the SRC assay data shown in original Fig. 2 and original Table S2. As shown in revised Fig. S8a (new data), both the CD34⁺ and CD34⁻ SRCs showed multi-lineage differentiation potentials, including CD34, CD33, CD19, CD14, CD11b, CD41, CD235a, CD3, CD4, and CD8, in the BM, PB, spleen and thymus in primary transplantation. As clearly seen in this figure, the repopulation level of CD235a in the BM (left tibia and other bone) of CD34⁻ SRCs is significantly higher than those of CD34⁺ SRCs. These findings are consistent with our *in vitro* CFC assay data shown in revised Figs. 1i and j (original Figs. 1i and j). We have now described these findings on page 8, line 27 to page 9, line 15.

2. *It would be important to assess erythroid output using a more sensitive system such as ex vivo culture of bone marrow xenograft (Hayakawa, Cell Transplant, 2010), NOD/SCID mice pretreated with clodronate or holo-transferrin, or by using NSGW41 mice or mice expressing human IL3 and EPO. If CD34⁻ cells do not possess E-Mk potential in vivo, it would indicate that CD34⁻ population include E-Mk CFCs without in vivo repopulation potential.*

Response: The reviewer's comment is reasonable. We therefore performed additional quantitative analyses of the multi-lineage differentiation potentials of CD34⁺ and CD34⁻

SRCs using the SRC assay data shown in original Fig. 2 and original Table S2. As shown in revised Fig. S8a (new data), both the CD34⁺ and CD34⁻ SRCs showed multi-lineage differentiation potentials, including CD34, CD33, CD19, CD14, CD11b, CD41, CD235a, CD3, CD4, and CD8, in the BM, PB, spleen and thymus in primary transplantation. As clearly seen in this figure, the repopulation level of CD235a in the BM (left tibia and other bone) and the repopulation level of CD41 in the BM (left tibia) and spleen of CD34⁻ SRCs are significantly higher than those of CD34⁺ SRCs. These findings are consistent with our *in vitro* CFC assay data shown in revised Figs. 1i and j (original Figs. 1i and j). We have now described these findings on page 8, line 27 to page 9, line 15. In addition, CD34⁻ SRCs showed primary, secondary and tertiary repopulating capacities by a single-cell-initiated serial transplantation analysis, as described on page 9, line 25 to page 11, line 22. These results demonstrated that CD34⁻ SRCs have potent *in vivo* repopulation potential.

3. *Authors should define the threshold for positive engraftment. Authors claim that CD34⁻ cells resorted from primary recipient show multilineage engraftment in secondary recipient. However, chimerism level obtained in secondary recipient is 0.01-0.02% (Table S2). This is an extremely low level (much lower than for CD34⁺ cells). I am not sure if this level is reliable and whether multilineage analysis at such a low level of engraftment is valid. Thus, the conclusion made by the authors that CD34⁻ HSCs cells are able to self-renew remains unproven. It is possible, that CD34⁻ cells have a limited self-renewal potential and represent a distinct type of HSCs with limited self-renewal potential, rather than HSCs that reside at the apex of human HSC hierarchy as depicted in Fig. 6b.*

Response: The reviewer's comment is quite reasonable. As previously reported (Leukemia 28:1308-1315,2014), mice were scored as positive if more than 0.01% of the total murine BM cells were human CD45⁺ cells. In separate experiments, we confirmed the detection limit of human CD45⁺ cells in mouse BMs by FCM was 0.005%. In order to address the reviewer's comment, we made a new figure (revised Fig. S7). In this figure, one representative primary recipient NOG mouse that received 18Lin⁻CD34⁻CD133⁺GPI-80⁺ cells is shown. We resorted 18Lin^{low/-}CD34⁺ and 18Lin^{low/-}CD34⁻ cells by FACS and these cells were transplanted to each secondary recipient mouse. After 19 to 23 weeks, these mice were euthanized, and the human CD45⁺ cell repopulation was analyzed by FCM. The mouse that received 18Lin^{low/-}CD34⁻ cells was repopulated with human CD45⁺ cells. However, the repopulation rate was 0.02 % in this representative mouse. As clearly seen in this revised Fig. S7c, we were able to detect human CD45⁺ cells with multi-lineage differentiation (CD33⁺ and CD19⁺ cells). Based on these results, we suggest that CD34⁻ SRCs were able to self-renew in NOG mice and that CD34⁻ HSCs reside at the apex of the human HSC hierarchy. We

have now described these findings on page 8, lines 20 to 24.

In this revised version, we added the data of single-cell-initiated tertiary transplantation analysis, as shown in revised Table S3 and revised Figs. 9a to d. Single CD34⁺ SRCs (HSCs) were able to engraft primary, secondary and tertiary recipient mice for over one year. These results strongly indicate that CD34⁺ SRCs (HSCs) are multipotent long-term repopulating HSCs with extensive self-renewing capacity. Taken together, these results support our present proposal shown in revised Fig. 7 (original Fig. 6). We have now described these findings on page 11, lines 12 to 22.

4. *In engraftment analysis, a sizable proportion of hCD45⁺ cells are positive for mCD45. Why are hCD45⁺mCD45⁺ cells are included in the analysis and why do the authors think that these are human cells and not mouse cells?*

Response: The reviewer's comment is quite reasonable. To address this comment, we performed an additional experiment. In this experiment, we euthanized one NSG female mouse (8 weeks old) that had not received human hematopoietic cell transplantation and collected the BM, PB and spleen cells. As a positive control, we used human CB-derived mononuclear cells (MNCs). All four cell populations were stained with anti-mouse CD45.1 mAb, anti-human CD45, anti-human CD34, anti-human CD33, and anti-human CD19 mAbs. As shown in Appendix Figs. 4a to c, most of the cells were detected in the mouse CD45.1-positive R1 gate. However, we did not detect any positive cells in the human CD45-positive R2 gate in the murine BM, PB, or spleen cells. As shown in Appendix Fig. 4d, we were able to detect human cells in this R2 gate clearly using human CB-derived MNCs. In contrast, we did not detect any mouse CD45.1-positive cells in the R1 gate. These results demonstrated that anti-mouse CD45.1 mAb did cross-react with human CD45⁺ cells. Conversely, anti-human CD45 mAb did not cross-react with mouse CD45.1⁺ cells. Based on these data, we designated hCD45⁺mCD45.1⁺ cells residing in the R2 gate as human hematopoietic cells. We have now described these findings on page 27, line 25 to page 28, line 3.

Appendix Figure 4.

5. *On page 9 “To provide an independent line of evidence for characterizing our highly purified CD34+/- SRCs (HSCs), we analyzed gene expression profiles of these two classes of SRCs (HSCs) at the single-cell level.” They should add that other types of progenitors and mature cells were analyzed as well. Otherwise, description of cluster analysis becomes confusing.*

Response: As suggested, we added the text “and other types of progenitors and mature cells” on page 12, line 1 in this revised version.

Additional corrections and explanations for the revision made:

- 1) In the revised version, we performed several additional experiments and analyses to address criticisms raised by the three reviewers. Based on the results of several additional experiments and analyses, we changed (added more data) original Figures 1, S5 and S8 and original Tables S2, S3 and S6. We also created new figures as revised Figures S1, S7, S8a and b and S10 based on the additional experiments and analyses. According to these revisions, the original figure numbers were changed as follows;
Original Figure 1 to revised Figure 1 with modifications.
Original Figure 2 to revised Figure 2.
Original Figure 3 to revised Figure 3.
Original Figure 4 to revised Figure 4
Original Figure 5 to revised Figure 5.
Original Figure S8 to revised Figure 6 with modifications.
Original Figure 6 to revised Figure 7.
Original Figure S1 to revised Figure S2.
Original Figure S2 to revised Figure S3.
Original Figure S3 to revised Figure S5.
Original Figure S4 to revised Figure S6.
Original Figures S5a to c to revised Figures S9a to d with modifications.
Original Figure S6 to revised Figure S12.
Original Figure S7 to revised Figure S4.
Original Figure S9 to revised Figure S11.

Overall, our revised version contains 7 figures, 12 supplemental figures, and 6 supplemental tables.

- 2) We corrected the description “CD34^{+/-}” to “CD34⁺ and CD34⁻” throughout the text, figures, figure legends and supplementary information according to comment 1 of Reviewer #1 .
- 3) In the original text, we mainly discussed the global gene expression analysis of CD34⁺ and CD34⁻ HSCs in the supplementary information. This was because according the post provisions of Nature Medicine, to which we first submitted this manuscript, stated that the maximum text length was 3,000 words. However, our manuscript is now being submitted to your journal, and the maximum text length is 5,000 words. We have therefore moved this text to the Result section on page 13, line 4 to page 15, line 6 in the revised version as suggested in other comment 6 of Reviewer #1.
- 4) In addition, we described the cell cycle status of CD34⁺ and CD34⁻ HSCs more precisely

in this revised version. We therefore moved the text concerning the cell cycle status (single-cell proliferation assay) of CD34⁺ and CD34⁻ HSCs, which was originally presented in the supplementary information, to the Result section of the main text on page 6, line 19 to page 7, line 11.

- 5) As suggested in comment 5 of Reviewer #1, we performed additional quantitative analyses of data shown in original Fig. 2 and original Table S2. Based on the findings of these new analyses, we have slightly revised the percentages of human CD45⁺ cells in the primary and secondary recipient mice, as shown in revised Table S2. During these analyses, we noticed some mistakes in the percentages of human CD45⁺ cells in the secondary recipient mice that received resorted CD34⁺ and CD34⁻ cells. The original data used the repopulation rates in other bones. However, as described in the footnote, we intended to present the repopulation rates of the injected site, e.g. both tibiae for secondary recipient mice. We have therefore corrected these data as described in the footnote.
- 6) In revised Table S3 (original Table S3), we added the data of the incidences of engraftment in the secondary recipient mice. In addition, we presented the precise data of single cell-initiated tertiary transplantation analyses.
- 7) In revised Table S6 (original Table S6), we added the five mAbs used for the analyses of the expression of CD34, CD49f, CD90, CD93 and CD184 (CXCR4) on our highly purified 18Lin⁻CD34⁺CD38⁻CD133⁺GPI-80⁺ and 18Lin⁻CD34⁻CD133⁺GPI-80⁺ cells.
- 8) In the Methods section, we added the experimental procedures for the analyses of the surface expression of CD49f, CD90, CD93 and CD184 (CXCR4) on our highly purified 18Lin⁻CD34⁺CD38⁻CD133⁺GPI-80⁺ and 18Lin⁻CD34⁻CD133⁺GPI-80⁺ cells on page 22, lines 13 to 25. In addition, we explained the experimental procedures for the secondary and tertiary transplantation on page 29, line 24 to page 30, line 24. We also described the method used for the statistical analysis of the difference in the colony-forming efficiency shown in revised Fig. 1i on page 34, lines 9 to 11.
- 9) In the original manuscript submitted to Nat Med and transferred to Nat Commun, we quoted 50 references in the main text and 16 references in the supplementary information. In this revised version, we quoted 58 references in the main text and 9 references in the supplementary information. References that have not been changed are printed in black. References newly added are printed in red. References that had their number changed are printed in blue.

Reviewers' comments:

Reviewer #1 (Remarks to the Author):

General Comments.

The revised paper by Sumide et. al. from the Sonoda group have addressed some of the minor concerns raised in the initial review. While it is clear that an enormous amount of work went into this paper, and it is possible that the authors are correct, it is not clear that the two main aims have been achieved. The authors suggest that the CD34⁻ population that they defined lies at the apex of hematopoiesis and before the CD34⁺ population in human cord blood, and that there is a by-pass "shunt" for these cells that produces erythroid megakaryocytic lineages. For the apex issue, why are these CD34⁻ cells not found in human bone marrow, if they are at the apex? This is a current issue of concern. For the by-pass issue, does this mean that there is a way that this process does not go through the CD34⁺ cell population, and if so, how?

In short, it is still not terribly convincing that CD34⁻CD133⁺GPI80⁺ cells (referred to as CD34⁻ SRCs/HSCs) localize at the apex of the hematopoietic hierarchy, especially compared with previously well-defined Lin⁻CD34⁺CD38⁻CD45RA⁻CD90⁺CD49f⁺ phenotypic HSC (See comment #1 below in Specific Comment Section). Also, there is no evidence to prove the reliability of the "bypass route" in the model they proposed. Since these are very important concepts, the proofs should be rigorous.

Specific Comments (Major).

1. The authors clarified that CD34⁻CD133⁺GPI80⁺ cells and CD34⁺CD38⁻CD133⁺GPI80⁺ cells are mostly CD49⁺ and CD90⁺, but these data were from 17 pooled cord blood samples. Due to the different genetic backgrounds of different cords, the expression levels of CD49 and CD90 could be quite different. They need to perform this experiment using a number of single cord blood units. Also since the frequency of CD34⁻CD133⁺GPI80⁺ cells is as low as 1/5,253,326, the gating they show in Fig. S1, which was based on the gated CD34⁻CD133⁺GPI80⁺ cells, does not look convincing. How many CD34⁻CD133⁺GPI80⁺ events were collected?

2. If the authors do not themselves perform cell transplantation, and preferably single cell transplantation, to compare CD34⁻CD133⁺GPI80⁺ HSCs and previously well defined Lin⁻CD34⁺CD38⁻CD45RA⁻CD90⁺CD49f⁺ HSCs, it's still hard to say which one is at apex. They cannot simply compare the SRCs in another group's study with the data shown in this manuscript. There must be an experiment in the author's lab that does a side by side comparison of the two rigorously defined HSC phenotypes with a least secondary transplants of these engrafted cells performed.

The authors claimed that the engraftment of human CD45 positive cell chimerism between CD34⁻CD133⁺GPI80⁺ cells and CD34⁺CD38⁻CD133⁺GPI80⁺ cells did not show significant differences. Does this mean there are two kinds of long-term HSCs in cord blood. Why are

these CD34⁻ cells not found in human bone marrow? Collection of CD34 “negative” cells may be caused by failed staining of CD34⁺ cells, especially since they mentioned that the CD34⁻ cells they collected included both CD34 mRNA –positive and –negative cells. Are the CD34⁻ and CD34⁺ cells just the same type of cells that transiently move back and forth even though they seem to have some different molecular characteristics?

3. CD34-SRCs showed a significantly higher megakaryocyte/erythrocyte lineage differentiation. From this, how did they come to the conclusion that CD34⁻ HSCs can directly differentiate into MEPs through a “bypass route”, as it is assumed that they do not go through the CD34⁺ cell population?

4. In the original review the question was raised as to whether Lin-CD34-CD133+GPI-80+ HSCs could be ex vivo expanded to give rise to more of these same stem cells or to CD34⁺ cells using SCF, TPO, Flt3L plus SR1 or UM171. This is a simple ex-vivo experiment. It would not have taken much time to do. However, the authors did not want to test this and planned to do it in their next study. This is important and if the experiment worked, it could further support their conclusions regarding the apex issue

5. The authors are still showing representative dot-blot results of an “n” of one. While the blots are nice, there is large variations noted between cord bloods, so it is imperative that all the data also be shown as a mean with error bars and statistics for all the numbers so that we can tell if these dot-blot results are truly representative.

Minor Comment.

The authors showed that human BM do not have Lin-CD34-CD133+ HSCs and may only contain CD34⁺ HSCs. If this is the case, Lin-CD34-CD133+GPI-80+ HSCs are only cord blood specific population. This should be emphasized. These Lin-CD34-CD133+GPI-80+ HSCs could be just transiently existing cell populations or they might be an intermediate population during development.

Reviewer #2 (Remarks to the Author):

I am satisfied with the revision and suggest acceptance in its present form.

Reviewer #3 (Remarks to the Author):

All my concerns are adequately addressed.

Authors should ensure consistency in terminology throughout the text. In some places (page 8) 18Linlow/- is used. I assume it should be 18Lin-.

RESPONSE TO REVIEWER 1.

We thank the reviewer for these critical but very constructive comments on our manuscript.

Specific comments (Major):

- 1. The authors clarified that CD34-CD133+GPI80+ cells and CD34+CD38-CD133+GPI80+ cells are mostly CD49+ and CD90+, but these data were from 17 pooled cord blood samples. Due to the different genetic backgrounds of different cords, the expression levels of CD49 and CD90 could be quite different. They need to perform this experiment using a number of single cord blood units. Also since the frequency of CD34-CD133+GPI80+ cells is as low as 1/5,253,326, the gating they show in Fig. S1, which was based on the gated CD34-CD133+GPI80+ cells, does not look convincing. How many CD34-CD133+GPI80+ events were collected?*

Response: The reviewer's comment is quite reasonable. As suggested, we performed additional experiments to analyze the expression of CD90 and CD49f on 18Lin⁻CD34⁺CD38⁻CD133⁺GPI-80⁺ and 18Lin⁻CD34⁻CD133⁺GPI-80⁺ cells derived from 7 single CB units. As shown in revised Figure S1 (representative data), most of the 18Lin⁻CD34⁺CD38⁻CD133⁺GPI-80⁺ and 18Lin⁻CD34⁻CD133⁺GPI-80⁺ cells expressed CD90 and CD49f, respectively. In addition, we simultaneously analyzed the expression of CD93 and CXCR4 on 18Lin⁻CD34⁺CD38⁻CD133⁺GPI-80⁺ and 18Lin⁻CD34⁻CD133⁺GPI-80⁺ cells derived from 7 single CB units. These results are comparable to the previous data obtained from 17 pooled CB units (original Figure S1). Precise data (range and mean ± S.D.) of the proportions of positive fractions of CD90, CD49f, CD93 and CXCR4 on 18Lin⁻CD34⁺CD38⁻CD133⁺GPI-80⁺ and 18Lin⁻CD34⁻CD133⁺GPI-80⁺ cells derived from 7 single CB units are depicted in the figure. The numbers of 18Lin⁻CD34⁺CD38⁻CD133⁺GPI-80⁺ and 18Lin⁻CD34⁻CD133⁺GPI-80⁺ cells (events) analyzed are described in the figure legend. We have now described these findings on page 5, lines 13 to 18.

- 2. If the authors do not themselves perform cell transplantation, and preferably single cell transplantation, to compare CD34-CD133+GPI80+ HSCs and previously well defined Lin-CD34+CD38-CD45RA-CD90+CD49f+ HSCs, it's still hard to say which one is at apex. They cannot simply compare the SRCs in another group's study with the data shown in this manuscript. There must be an experiment in the author's lab that does a side by side comparison of the two rigorously defined HSC phenotypes with a least secondary transplants of these engrafted cells performed.*

Response: Our identified CD34⁻ SRCs (HSCs) (18Lin⁻CD34⁻CD133⁺GPI-80⁺ cells) were able to generate CD34⁺ SRCs (HSCs) both *in vitro* (Stem Cells, 2007; Stem Cells, 2015; Blood, 2016) as well as *in vivo* (Blood, 2003; Cell Transplant, 2017), and as shown *in vivo* in this study (revised Figure S7). Furthermore, most of the 18Lin⁻CD34⁺CD38⁻CD133⁺GPI-80⁺ cells (CD34⁺ HSCs) expressed CD90 and CD49f, as shown in revised Figure S1. These results suggested that our identified CD34⁺ HSCs (18Lin⁻CD34⁺CD38⁻CD133⁺GPI-80⁺ cells) were a very close population to the previously reported CD34⁺ HSCs (9Lin⁻CD34⁺CD38⁻CD45RA⁻CD90⁺CD49f⁺ cells). In this study, single-cell-initiated serial transplantation analyses clearly demonstrated that both CD34⁻ HSCs (18Lin⁻CD34⁻CD133⁺GPI-80⁺ cells) and CD34⁺ HSCs (18Lin⁻CD34⁺CD38⁻CD133⁺GPI-80⁺ cells) showed primary, secondary and tertiary repopulating capacities. These results indicated that both CD34⁺ and CD34⁻ HSCs were LT-HSCs. It was therefore very difficult to observe the difference in the LT-repopulating capacity of these CD34⁺ and CD34⁻ HSCs, even after tertiary transplantation.

In this comment, the reviewer asked us which one was at apex of the human HSC hierarchy. To address this important question, we performed new experiments. We co-cultured 18Lin⁻CD34⁻CD133⁺GPI-80⁺ cells (CD34⁻ HSCs) and 9Lin⁻CD34⁺CD38⁻CD45RA⁻CD90⁺CD49f⁺ cells (CD34⁺ HSCs) with DP MSCs (Stem Cells, 2015). As shown in revised Figure S10 and Table S4, 18Lin⁻CD34⁻CD133⁺GPI-80⁺ cells (CD34⁻ HSCs) were able to produce 9Lin⁻CD34⁺CD38⁻CD45RA⁻CD90⁺CD49f⁺ cells with SRC activity (CD34⁺ HSCs) *in vitro*. Furthermore, CD34⁻ HSCs were able to produce CD34⁻ SRCs (HSCs) *in vitro* as shown *in vivo* (Figure S7). In contrast, 9Lin⁻CD34⁺CD38⁻CD45RA⁻CD90⁺CD49f⁺ cells (CD34⁺ HSCs) were only able to produce the same class of CD34⁺ HSCs. They were not able to produce CD34⁻ SRCs (HSCs) *in vitro*. These results suggested that CD34⁻ SRCs (HSCs) were more immature than CD34⁺ SRCs (HSCs), placing them at the apex of the human CB-derived HSC hierarchy. We have now described these findings on page 11, line 8 to page 12, line 6.

The reviewer's comment (continued)

The authors claimed that the engraftment of human CD45 positive cell chimerism between CD34-CD133+GPI80+ cells and CD34+CD38-CD133+GPI80+ cells did not show significant differences. Does this mean there are two kinds of long-term HSCs in cord blood.

Response: As mentioned above, both CD34⁻ HSCs (18Lin⁻CD34⁻CD133⁺GPI-80⁺ cells) and CD34⁺ HSCs (18Lin⁻CD34⁺CD38⁻CD133⁺GPI-80⁺ cells) showed single-cell-initiated primary, secondary and tertiary repopulating capacities. These results indicated that both CD34⁺ and CD34⁻ HSCs were LT-HSCs. We therefore believe that there are two kinds of LT-HSCs in human CB. We have now described these findings on page 16, lines 17 to 20.

The reviewer's comment (continued)

Why are these CD34⁻ cells not found in human bone marrow?

Response: As mentioned, we did not detect 18Lin⁻CD34⁻CD133⁺GPI-80⁺ cells in the BM samples, as shown in Figure S14. However, we used middle-aged volunteer donor-derived BM cells. Therefore, 18Lin⁻CD34⁻CD133⁺GPI-80⁺ cells may be present in the BM cells of young adults or children. Alternatively, these results may suggest that CD34⁻ HSCs (18Lin⁻CD34⁻CD133⁺GPI-80⁺ cells) are a CB-specific population, as suggested by the minor comment of this reviewer. We have therefore described these findings and their potential implications on page 20, lines 8 to 11.

The reviewer's comment (continued)

Collection of CD34 “negative” cells may be caused by failed staining of CD34⁺ cells, especially since they mentioned that the CD34⁻ cells they collected included both CD34 mRNA –positive and –negative cells. Are the CD34⁻ and CD34⁺ cells just the same type of cells that transiently move back and forth even though they seem to have some different molecular characteristics?

Response: We believe that our immuno-staining method was appropriate, as we were able to detect CD34⁺ and CD34⁻ cells by FACS. The detection of CD34 antigen expressed on the cell surface by FACS depends on the protein expression. However, there are limits to detection with FACS technology. Therefore, we were unable to detect the small amount of CD34 antigen expressed on the cell surface. As a result, these cells were evaluated as CD34⁻ cells by FACS despite containing CD34 mRNA. Given the above, it is no wonder that the FACS-sorted CD34⁻ cells contained CD34 mRNA-positive and CD34 mRNA-negative cells. As shown in Figure 5c(ii), the expression of CD34 mRNA in FACS-sorted CD34⁺ cells was significantly higher (Student's *t*-test, $P < 0.001$) than that of the CD34 mRNA⁺ cells in the FACS-sorted CD34⁻ cells. We then compared the expression of CD34 mRNAs between FACS-sorted CD34⁻ cells and CD34⁺ cells by a microarray analysis. As a result, we confirmed that the expression of CD34 mRNA of FACS-sorted CD34⁻ cells was significantly lower (Student's *t*-test, $P < 0.001$) than that of FACS-sorted CD34⁺ cells, as shown in Appendix Figure 1.

Appendix Figure 1.

The gene expression profiles of 18Lin⁻CD34⁺CD38⁻CD133⁺GPI-80⁺ cells (CD34⁺ HSCs) and 18Lin⁻CD34⁻CD133⁺GPI-80⁺ cells (CD34⁻ HSCs) were clearly different, as shown in Figures 4 and 5 (single-cell gene expression analyses) and Figures 6 and S11 (microarray analyses). In addition, we found that the CD34⁻ HSCs showed potent megakaryocyte/erythrocyte differentiation potential *in vitro* (Figures 1i and j) and *in vivo* (Figure S8a). These findings are consistent with the microarray analysis result that CD34⁻ HSCs expressed higher levels of gene sets related to megakaryocyte/erythrocyte differentiation (Figure 6). These results demonstrated that there was a clear functional difference between CD34⁺ and CD34⁻ SRCs (HSCs). Furthermore, CD34⁻ SRCs (HSCs) were able to generate CD34⁺ SRCs (HSCs) *in vitro* and *in vivo*, as reported previously and as shown in this study (Figure S10, Table S4 and Figure S7). However, it is hard to prove that CD34⁺ SRCs (HSCs) were able to produce CD34⁻ SRCs (HSCs) *in vitro* and *in vivo*. Given these findings, we believe that these CD34⁺ and CD34⁻ HSCs are a different class of human HSCs.

3. CD34⁻ SRCs showed a significantly higher megakaryocyte/erythrocyte lineage differentiation. From this, how did they come to the conclusion that CD34⁻ HSCs can directly differentiate into MEPs through a “bypass route”, as it is assumed that they do not go through the CD34⁺ cell population?

Response: As shown in Figures 1i and j, CD34⁻ HSCs showed a high incidence of CFU-EM colony formation. In addition, CD34⁻ SRCs (HSCs) showed significantly higher levels of CD41⁺ and CD235a⁺ cell repopulations than did CD34⁺ SRCs (HSCs) (Figure S8a). Furthermore, CD34⁻ HSCs expressed a higher level of gene sets related to megakaryocyte/erythrocyte differentiation (Figure 6). Based on these findings, we

concluded that CD34⁻ SRCs (HSCs) possessed higher megakaryocyte/erythrocyte differentiation potential than CD34⁺ SRCs (HSCs). *In vivo* repopulation analyses clearly demonstrated that CD34⁻ SRCs (HSCs) showed multi-lineage differentiation potential (Figures 2, 3, S6, S7, S8 and S9, and Tables S2 and S3). Furthermore, CD34⁻ SRCs (HSCs) generated CD34⁺ SRCs (HSCs) *in vivo* (Figure S7) and *in vitro* (Figure S10 and Table S4). Given these findings, we believe that CD34⁻ HSCs first differentiate to CD34⁺ HSCs and then to MPP, CMP, MLP and so on in line with the proposed model (Figure 7b) of steady-state hematopoiesis. However, as shown in Figures 1i and 1j, CD34⁻ HSCs produced large numbers of CFU-EM colonies *in vitro*. Therefore, we believe that CD34⁻ HSCs may directly differentiate to MEP in methylcellulose culture, in which niche cells/factors are absent. We also imagine that CD34⁻ HSCs may express their potent megakaryocyte/erythrocyte differentiation potential *in vivo*, especially when they are placed in an environment less affected by niche cells/factors. Based on these findings and our own inference, we propose a bypass route. We have now described these findings on page 19, line 22 to page 20, line 7.

4. *In the original review the question was raised as to whether Lin-CD34-CD133+GPI-80+ HSCs could be ex vivo expanded to give rise to more of these same stem cells or to CD34+ cells using SCF, TPO, Flt3L plus SR1 or UM171. This is a simple ex-vivo experiment. It would not have taken much time to do. However, the authors did not want to test this and planned to do it in their next study. This is important and if the experiment worked, it could further support their conclusions regarding the apex issue*

Response: As suggested, we performed additional experiments in order to investigate the effects of SR-1 or UM171 on the *ex vivo* expansion of CD34⁻ SRCs (HSCs). We tried to expand very primitive CB-derived CD34⁻ SRCs (HSCs) (18Lin⁻CD34⁻CD133⁺GPI-80⁺ cells) using SCF+TPO+FL+SR-1 or UM171. As shown in Figures S15c and d, 18Lin⁻CD34⁻CD133⁺GPI-80⁺ cells actively proliferated and showed 460-fold (UM171) to 950-fold (SR-1) increase of total numbers of cells, yielding 9 x 10⁵ (UM171) to 13 x 10⁵ (SR-1) 12Lin⁻CD45RA⁻CD34⁺ cells. We then performed LDAs to analyze their effects on the expansion of SRCs (Figures S15e and f). Unfortunately, we were unable to expand CD34⁻ SRCs (HSCs). As shown in this study (Figures S7 and S10), CD34⁻ SRCs (HSCs) did generate CD34⁺ SRCs (HSCs) in these culture conditions in the presence of cytokines plus SR-1 or UM171. However, the expansion efficiencies of SRCs were 1.33 for SR-1 and 0.42 for UM171. These efficiencies were not statistically significant in comparison to DMSO (control). These results showed that the *ex vivo* expansion of primitive CB-derived CD34⁻ SRCs (HSCs) was more difficult than previously considered, suggesting that *ex vivo* expansion of primitive CB-derived CD34⁻ SRCs (HSCs) may require unidentified niche cells/factors for their maintenance and/or proliferation. Further studies will be

needed to clarify the details of this important issue. We have now described these findings on page 20, lines 14 to 24.

5. *The authors are still showing representative dot-blot plots of an “n” of one. While the plots are nice, there is large variation noted between cord bloods, so it is imperative that all the data also be shown as a mean with error bars and statistics for all the numbers so that we can tell if these dot-blot results are truly representative.*

Response: As pointed out, we showed representative dot-blot plots in original Figures 1a to 1f, 2d and 2e, 3a and 3b, S1, S6, and S7. In this revised version, we now share the detailed FCM data, including numbers of experiments, mean \pm S.D. and ranges of proportions of cells in each gate, in the figures or figure legends of Figures 1, 2 and 3, and Figures S1, S6 and S7, and Tables S2, and Tables S3b to d. In the original Figure 1a to 1f, we presented one FACS result as representative of 32 experiments. However, the proportions of cells in R6 and R8 gates were near the lower range. We have therefore changed this FACS result to another one which we feel more appropriate (near mean value) as a representative finding.

Minor Comment.

The authors showed that human BM do not have Lin-CD34-CD133+ HSCs and may only contain CD34+ HSCs. If this is the case, Lin-CD34-CD133+GPI-80+ HSCs are only cord blood specific population. This should be emphasized. These Lin-CD34-CD133+GPI-80+ HSCs could be just transiently existing cell populations or they might be an intermediate population during development.

Response: As mentioned above, we did not detect 18Lin⁻CD34⁻CD133⁺GPI-80⁺ cells in the BM samples, as shown in Figure S14. However, we used middle-aged volunteer donor-derived BM cells. Therefore, 18Lin⁻CD34⁻CD133⁺GPI-80⁺ cells may be present in the BM cells of young adults or children. Alternatively, as suggested by the reviewer, CD34⁻ HSCs (18Lin⁻CD34⁻CD133⁺GPI-80⁺ cells) may be a CB-specific population. We have now described this possibility on page 20, lines 8 to 11, in the revised version.

RESPONSE TO REVIEWER 2.

We thank the reviewer for their evaluation and acceptance of our manuscript.

Reviewer #2 (Remarks to the Author):

I am satisfied with the revision and suggest acceptance in its present form.

RESPONSE TO REVIEWER 3.

We thank the reviewer for their evaluation of our manuscript and constructive comments on our manuscript.

Reviewer #3 (Remarks to the Author):

All my concerns are adequately addressed.

Authors should ensure consistency in terminology throughout the text. In some places (page 8) 18Linlow/- is used. I assume it should be 18Lin-.

Response: As the reviewer suggested, we have now corrected “18Lin^{low/-}” to “18Lin-” on page 7, lines 21, 24, 25, and page 8, lines 2 and 4.

Additional corrections and explanations for the revision made:

- 1) In the revised version, we performed several additional experiments and analyses to address criticisms raised by reviewer #1. Based on the results of several additional experiments and analyses, we changed (added more data) original Figures 1, S3, S7, and S8 and original Tables S2, S3 and S6 (revised Table S5). We also created new figures and tables as revised Figures S1, S10, S14, and S15, and Tables S2b, c and d, S3b, c and d, and S4 based on the additional experiments and analyses. According to these revisions, the original figure and table numbers were changed as follows;

Original Figure 1 to revised Figure 1 with modifications.

Original Figure 2 to revised Figure 2 with correction.

Original Figure 3 to revised Figure 3 without modification.

Original Figure 4 to revised Figure 4 without modification.

Original Figure 5 to revised Figure 5 without modification.

Original Figure 6 to revised Figure 6 without modification.

Original Figure 7 to revised Figure 7 without modification.

Original Figure S1 to revised Figure S1 which was newly created based on the additional experiments.

Original Figure S2 to revised Figure S2 without modification.

Original Figure S3 to revised Figure S3 with modification. We added the number of experiments.

Original Figure S4 to revised Figure S4 without modification.

Original Figure S5 to revised Figure S5 without modification.

Original Figure S6 to revised Figure S6 without modification.

Original Figure S7 to revised Figure S7 with modification. We added precise data of secondary transplantation of CD34⁺ SRCs in addition to CD34⁻ SRCs. And statistical analysis data (range and mean \pm S.D.) are also depicted in the figure.

Original Figure S8a(i) to revised Figure S8a(i) with correction.

Original Figure S8b to revised Figure S8b without modification.

Original Figures S9a to d to revised Figures S9a to d without modification.

Revised Figure S10 was newly created based on the additional experiments.

Original Figure S10 to revised Figure S11 without modification.

Original Figure S11 to revised Figure S12 without modification.

Original Figure S12 to revised Figure S13 without modification.

Revised Figure S14 was newly created based on the additional experiments.

Revised Figure S15 was newly created based on the additional experiments.

Original Table S1 to revised Table S1 without modification.

Original Table S2 to revised Tables S2a with modification. We added statistical analysis data (mean \pm S.D.).

Revised Tables S2b to d were created based on the multi-lineage analysis.

Original Table S3 to revised Tables S3a without modification.

Revised Table S3b to d were created based on the multi-lineage analysis.

Revised Table S4 was newly created based on the additional co-culture experiments.

Original Table S4 to revised Table S6 without modification.

Original Table S5 to revised Table S7 without modification.

Original Table S6 to revised Table S5 with modification. We added two antibodies used in the additional experiments.

Overall, our revised version contains 7 figures, 15 supplemental figures, and 7 supplemental tables. In addition, we created Appendix Figure 1 based on the microarray data to respond to comment 2 of reviewer #1.

- 2) In this revised version, we showed the detailed FCM data, including numbers of experiments, mean \pm S.D. and ranges of proportions of cells in each gate, in Figure 1. In the original Figure 1a to 1f, we presented one FACS data as a representative of 32 experiments. However, the proportions of cells in R6 and R8 gates were near the lower ranges. We have therefore changed this FACS result to another one which we feel more appropriate (near mean value) as a representative finding.
- 3) We corrected the Y scale bar of Figure 2b.
- 4) As suggested in comment 1 of reviewer #1, we performed additional experiments to analyze the expression of CD90, CD49f, CD93 and CXCR4 on 18Lin⁻CD34⁺CD38⁻CD133⁺GPI-80⁺ and 18Lin⁻CD34⁻CD133⁺GPI-80⁺ cells derived from 7 single CB units. Based on these experiments, we created new Figure S1.
- 5) In original Figure S8a(i), the dot plots of the PB and spleen were mixed up. We have now corrected this mistake.
- 6) In this revised version, we referenced five additional papers as Nos. 51 to 55 in red. We have also changed the No. of other reference papers based on these additions. Other references that had their numbers changed are indicated in red.
- 7) In the Supplementary Results and Discussion, we presented the data concerning the *ex vivo* expansion of CD34⁻ HSCs using TPO+SCF+FL plus SR-1 or UM171 in serum-free cultures.
- 8) In the Supplementary Methods section, we added the experimental procedures for the analyses of the presence of 18Lin⁻CD34⁺CD38⁻CD133⁺GPI-80⁺ and 18Lin⁻CD34⁻CD133⁺GPI-80⁺ cells in adult human BM cells. In addition, we added the experimental procedures for the co-cultures of 9Lin⁻CD34⁺CD38⁻CD45RA⁻CD90⁺CD49f⁺ cells (CD34⁺ HSCs) and

18Lin⁻CD34⁻CD133⁺ GPI-80⁺ cells (CD34⁻ HSCs) with human BM-derived DP MSCs. The *ex vivo* expansion procedures of 18Lin⁻CD34⁻CD133⁺ GPI-80⁺ cells (CD34⁻ HSCs) using cytokines plus SR-1 or UM171 were also presented.

- 9) We added several papers written in red in the supplementary references. We have also changed the No. of other reference papers based on these additions. Other references that had their numbers changed are also indicated in red.
- 10) In this revised version, we shortened the explanation of the cell cycle status of CD34⁺ and CD34⁻ HSCs, which appeared on page 6, lines 18 to 22, in order to fit the post-provision. These precise explanations have now been moved to the Supplementary Results and Discussion.
- 11) In this revised version, we explained the data of single-cell-initiated tertiary transplantation (Figure S9d) more precisely.

REVIEWERS' COMMENTS:

Reviewer #1 (Remarks to the Author):

General Comments

Thank you for responding so thoroughly to our comments. Your paper is now much improved however, it is extremely important that it becomes very clear in the title and abstract that at present the CD34- human HSCs have been shown to preside at the pinnacle of the road map only for cord blood.

Specific Comments (Minor, but important)

1. Title: Add the word cord blood after human and before CD34-negative.
2. To the abstract:
 - a. line 6: add: "from cord blood" after the word hierarchy
 - b. last line: add: "in cord blood" after the last word, HSCs.